# Molybdenum systematics of subducted crust record reactive fluid flow from underlying slab serpentine dehydration

Shuo Chen [1,2,3,4]*, Remco C. Hin[2], Timm John[5], Richard Brooker [2], Ben Bryan[2], Yaoling Niu[1,3,6] & Tim Elliott[2]

Fluids liberated from subducting slabs are critical in global geochemical cycles. We investigate the behaviour of Mo during slab dehydration using two suites of exhumed fragments of subducted, oceanic lithosphere. Our samples display a positive correlation of $\delta^{98/95}Mo_{NIST\ 3134}$ with Mo/Ce, from compositions close to typical mantle (−0.2‰ and 0.03, respectively) to very low values of both $\delta^{98/95}Mo_{NIST\ 3134}$ (−1‰) and Mo/Ce (0.002). Together with new, experimental data, we show that molybdenum isotopic fractionation is driven by preference of heavier Mo isotopes for a fluid phase over rutile, the dominant mineral host of Mo in eclogites. Moreover, the strongly perturbed $\delta^{98/95}Mo_{NIST\ 3134}$ and Mo/Ce of our samples requires that they experienced a large flux of oxidised fluid. This is consistent with channelised, reactive fluid flow through the subducted crust, following dehydration of the underlying, serpentinised slab mantle. The high $\delta^{98/95}Mo_{NIST\ 3134}$ of some arc lavas is the complement to this process.

---

[1] Key Laboratory of Marine Geology and Environment, Institute of Oceanology, Chinese Academy of Sciences, Qingdao 266071, China. [2] Bristol Isotope Group, School of Earth Sciences, University of Bristol, Bristol BS8 1RJ, UK. [3] Laboratory for Marine Geology, Qingdao National Laboratory for Marine Science and Technology, Qingdao 266061, China. [4] Center for Ocean Mega-Science, Chinese Academy of Sciences, 7 Nanhai Road, Qingdao 266071, China. [5] Institut für Geologische Wissenschaften, Freie Universität Berlin, Malteserstrasse 74-100, 12249 Berlin, Germany. [6] Department of Earth Sciences, Durham University, Durham DH1 3LE, UK. *email: shuochen@qdio.ac.cn

Subduction zones are major sites of chemical exchange between Earth's deep interior and surface. The subducting oceanic lithosphere releases fluids through a series of metamorphic, dehydration reactions which induce melting in the mantle wedge and result in arc magmatism. The magnitude and origin of these key fluid fluxes, however, remain uncertain[1–3]. The primary source of water to subduction zones has traditionally been linked to the breakdown of hydrous minerals contained in the oceanic crust[4]. However, more recent synthesis of high-pressure experiments, geochemical observations and geophysical models argue that dehydration of subducting serpentinite plays a more critical role in elemental transport and arc magma generation (see review in ref. [5]).

Whilst the importance for serpentine in bringing water down to sub-arc depths is now widely acknowledged, two contrasting scenarios are envisaged for its involvement[5]. In one endmember, serpentine forms above the slab, trapping fluids released from the crust in the shallow subduction zone, and is down-dragged with the descending plate. This serpentine potentially forms a mechanical melange with the underlying crust and ultimately rises diapirically[6]. This process has been invoked to account for the geology of some exhumed terrains[7], and aspects of arc magmatism[8]. A second conceptual model instead places the serpentine layer beneath the subducting crust, as a result of ingress of water along transform faults or during plate bending prior to subduction[9]. Serpentine dehydration beneath the subducting crust facilitates the removal of elements from the whole crustal section during subsequent passage of the fluid into the source of arc magmas and so this scenario has major ramifications for element cycling[10]. Although an increasing number of seismological studies have presented evidence for significant subduction of serpentinised slab mantle at various destructive plate margins[11–13], its importance has also been vigorously questioned[14]. New constraints to distinguish between these different models of serpentine involvement in arc magmatism would therefore be very valuable.

Molybdenum (Mo) systematics show promise for tracing the origin of fluids in subduction zones for two reasons. First, high-pressure experiments have shown that Mo mobility reflects the redox condition of slab-derived fluids[15]. Second, recent studies[16–19] have shown that arc lavas are generally enriched in Mo and have $\delta^{98/95}Mo_{NIST\ 3134}$ that are higher than values of the primitive mantle[20], depleted mid-ocean-ridge basalts[21] and most oceanic sediments[17]. The high $\delta^{98/95}Mo_{NIST\ 3134}$ in depleted arc lavas has been inferred to reflect Mo isotopic fractionation caused in its partitioning into fluids traversing the subducting crust from underlying, serpentine dehydration[17,18]. If correct, then Mo isotope systematics can identify the style of serpentine involvement in arc magmatism, although other interpretations of high $\delta^{98/95}Mo_{NIST\ 3134}$ in arc lavas have been proposed[16,19].

Here, by analysing material complementary to arc magmatism, namely exhumed fragments of subducted oceanic lithosphere, we provide a fresh perspective on the cause of Mo isotopic fractionation during subduction. In natural and experimental samples, we show that Mo in rutile, a common accessory phase in the mafic, subducted crust, is isotopically light. We further report low $\delta^{98/95}Mo_{NIST\ 3134}$ and Mo/Ce in bulk eclogite and blueschist samples that reflect the control of residual rutile during fractional loss of Mo caused by a large, oxidised, reactive-fluid flux.

## Results
**Samples**. We have examined samples from two well-characterized metaophiolite complexes (Table 1; Supplementary Table 1). The Raspas Complex, Ecuador, contains metasediments,

blueschists, eclogites, and eclogite-facies, serpentinised peridotite, which all experienced similar peak pressure-temperature conditions of ~ 2 GPa and $600 \pm 50\ °C$[22,23]. The Cabo Ortegal Complex eclogites record comparable peak metamorphic conditions of 650–670 °C, using Zr-in-rutile data from Schmidt et al.[24] in combination with a temperature calibration from Ferry and Watson[25] and pressures > 1.7 GPa[26].

The eclogites from both suites have Nd isotope ratios and key trace-element ratios like those of mid-ocean ridge basalts (MORB), see Fig. 1a. This indicates a depleted, oceanic crustal protolith, as expected of typical subduction zones and further precludes significant contributions from sediment melts or crustal contamination[23,27]. Although the Raspas blueschists are more 'enriched', they still have oceanic geochemical affinities (Fig. 1a) and are inferred to represent subducted seamounts[23]. Elements which show consistent, depleted, MORB-like characteristics in the eclogites (e.g. rare earth elements, high field strength elements and thorium) are empirically inferred to be fluid immobile[10,23] and this set of elements is consistent with those considered to be fluid immobile from arc lavas systematics[28]. More strictly, these observations suggest that there is residual zoisite and rutile along the pathway of percolating fluids which results in the immobility of these elements.

Fluid mobile elements reflect a more complex history (Fig. 1b). Compared with a MORB protolith, Sr is depleted relative to Nd, a fluid immobile element of similar magmatic incompatibility. Such elemental loss to slab-derived fluids is consistent with arc lava systematics[28,29]. Barium has also been inferred to be highly fluid mobile during subduction[28] and it has been observed to be specifically enriched in fluid inclusions within the Raspas eclogites[30]. Yet, relative to its fluid immobile analogue Th, Ba is not as ubiquitously depleted as Sr in the eclogites (Fig. 1b). The anomalously high concentrations of Ba (and some other elements such as Pb and K, not shown) can be linked to secondary formation of phengite from late percolating fluid, which can locally re-enrich eclogites in fluid mobile elements compatible in this phase[23].

**Molybdenum systematics of exhumed oceanic crust**. As with some of the more traditionally used fluid mobile elements (Fig. 1b), our samples show notable perturbations of their Mo elemental and isotope systematics relative to unaltered, depleted MORB (Fig. 2). Given comparably incompatible behaviour of Mo and Ce in magmatic processes, oceanic basalts have nearly constant Mo/Ce[31]. All the eclogite samples studied here have Mo/Ce considerably lower than this canonical mantle value of ~ 0.03[31]. The majority of eclogites also have significantly isotopically lighter Mo than either altered or unaltered oceanic crust (Fig. 2). There is a general trend from unaltered, depleted MORB towards low Mo/Ce and low $\delta^{98/95}Mo_{NIST\ 3134}$, albeit with two outliers with less negative $\delta^{98/95}Mo_{NIST\ 3134}$ at low Mo/Ce. Notably, the sample array appears to originate from a normal MORB composition rather than a significantly altered oceanic crustal (AOC) protolith (Fig. 2).

For one eclogite sample (SEC46-1) we have analysed the Mo concentrations and isotope compositions of individual phases (Table 1). In keeping with the work of Zack et al.[32], rutile is the dominant host of Mo (~ 5 μg g$^{-1}$). Rutile is also isotopically lighter ($\delta^{98/95}Mo_{NIST\ 3134} = -0.75‰$) than co-existing omphacite ($\delta^{98/95}Mo_{NIST\ 3134} = -0.16‰$) and garnet ($\delta^{98/95}Mo_{NIST\ 3134} = -0.29‰$). Compositional and modal data from the minerals can be used to reconstruct a bulk eclogite value ($\delta^{98/95}Mo_{NIST\ 3134} = -0.58‰$) within error of the measured whole rock analysis (Table 1). We further analysed rutile separates from one of the perturbed, higher Ba/Th samples (SEC43-1) which gave

**Table 1 $\delta^{98/95}Mo_{NIST\ 3134}$ and Mo concentrations in high-pressure mafic and ultra-mafic samples from Raspas Complex and Cabo Ortegal Complex**

| Sample | Rock/mineral type | [Mo] µg g$^{-1}$ | $\delta^{98/95}Mo_{NIST\ 3134}$ (‰) | n | Mo/Ce[a] | $^{143}Nd/^{144}Nd$[b] |
|---|---|---|---|---|---|---|
| *Raspas Complex (Ecuador)* | | | | | | |
| SEC15-2 | Blueschist | 0.383 | −0.49 ± 0.03 | 3 | 0.011 | |
| SEC16-1 | Blueschist | 0.102 | −0.99 ± 0.03 | 1 | 0.002 | 0.512870(3) |
| SEC42-6 | MORB-type eclogite | 0.178 | −0.38 ± 0.03 | 3 | 0.011 | 0.513072(2) |
| SEC43-1 | MORB-type eclogite | 0.052 | −0.37 ± 0.05 | 1 | 0.004 | 0.513174(2) |
| SEC43-1[c] | MORB-type eclogite | 0.055 | −0.35 ± 0.05 | 1 | 0.005 | |
| | Rutile | 2.440 | −0.31 ± 0.05 | 3 | | |
| SEC43-3 | MORB-type eclogite | 0.145 | −0.53 ± 0.03 | 1 | 0.011 | 0.513214(3) |
| SEC44-1 | MORB-type eclogite | 0.042 | −0.20 ± 0.04 | 1 | 0.004 | 0.513182(3) |
| SEC44-1[c] | MORB-type eclogite | 0.043 | −0.13 ± 0.04 | 1 | 0.004 | |
| SEC46-1 | MORB-type eclogite | 0.137 | −0.57 ± 0.04 | 1 | 0.010 | 0.513213(2) |
| | Fine rutile | 5.330 | −0.75 ± 0.08 | 3 | | |
| | Coarse rutile | 4.418 | −1.26 ± 0.02 | 3 | | |
| | Garnet | 0.064 | −0.29 ± 0.05 | 1 | | |
| | Omphacite | 0.141 | −0.16 ± 0.05 | 3 | | |
| SEC46-2 | MORB-type eclogite | 0.058 | −0.68 ± 0.04 | 1 | 0.004 | |
| SEC47-1 | MORB-type eclogite | 0.102 | −0.52 ± 0.03 | 2 | 0.009 | 0.513189(3) |
| SEC50-1 | Retrogressed eclogite | 0.130 | −0.45 ± 0.01 | 2 | 0.011 | |
| SEC26-3 | Serpentinised peridotite | 0.017 | 0.13 ± 0.10 | 1 | 0.515 | |
| SEC26-3[c] | Serpentinised peridotite | 0.016 | 0.14 ± 0.08 | 1 | 0.485 | |
| SEC35-2 | Serpentinised peridotite | 0.007 | | 1 | 0.500 | |
| *Cabo Ortegal Complex (Spain)* | | | | | | |
| SCO1-1 | MORB-type eclogite | 0.149 | −0.40 ± 0.01 | 2 | 0.016 | 0.513117(4) |
| SCO2-1 | MORB-type eclogite | 0.292 | −0.37 ± 0.05 | 3 | 0.018 | 0.512929(3) |
| SCO9-2 | MORB-type eclogite | 0.105 | −0.43 ± 0.03 | 1 | 0.017 | 0.513233(4) |
| SCO12-4 | MORB-type eclogite | 0.102 | −0.58 ± 0.04 | 1 | 0.012 | 0.513238(4) |
| SCO16-1 | MORB-type eclogite | 0.091 | −0.49 ± 0.03 | 1 | 0.015 | 0.513233(4) |
| SCO18-1 | MORB-type eclogite | 0.493 | −0.33 ± 0.02 | 3 | 0.031 | 0.513151(4) |
| SCO23-1 | MORB-type eclogite | 0.089 | −0.46 ± 0.04 | 1 | 0.013 | 0.513237(4) |

Uncertainties of $\delta^{98/95}Mo_{NIST\ 3134}$ are given as two standard deviation (2 SD, n > 1) or two standard error (2SE, n = 1), where n refers to number of isotopic analyses using the same solution
[a]Ce concentration is from refs. [23,40]
[b] $^{143}Nd/^{144}Nd$ of the eclogites from Cabo Ortegal are measured in this study (see Supplementary information); others are from ref. [27]; numbers in parentheses are two standard errors
[c]Replicate analysis on separate dissolution of the sample powder. On sample SEC46-1, we made two measurements of rutile separates: one picked very carefully to include only distinct, fine rutile needles (labelled fine rutile) and another less carefully picked that included larger, more irregular grains (labelled coarse rutile). Although both separates have notably isotopically light Mo, they are different and perhaps reflect different generations of growth from different fluid compositions. We have somewhat arbitrarily assumed equal proportions of the different phases in reconstituting the whole rock Mo isotope composition of SEC46-1 (−0.58‰ comparing to the measured −0.57 ± 0.04‰)

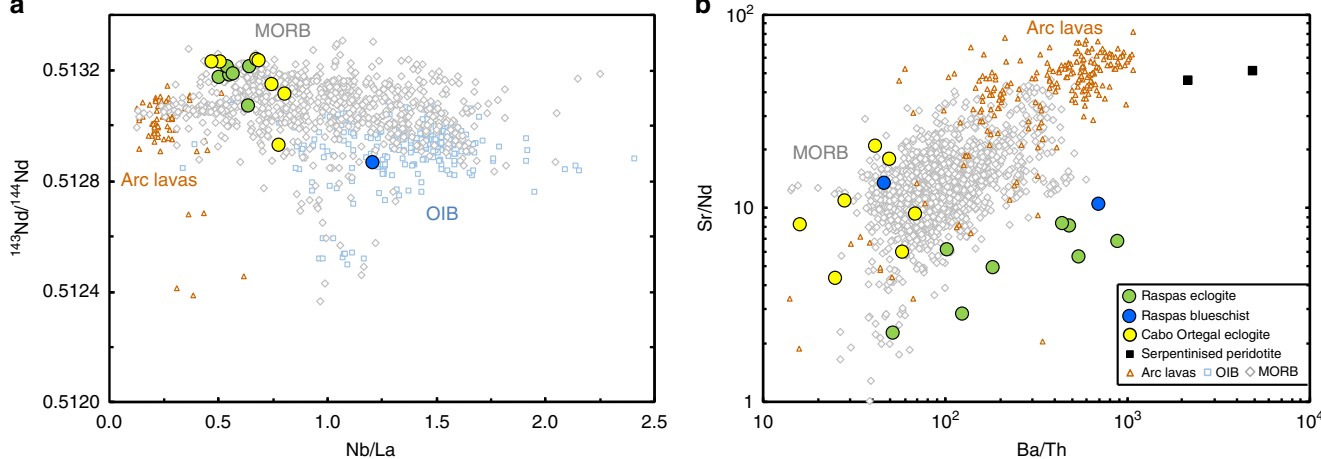

**Fig. 1** Protoliths and subduction modification of studied eclogites and blueschists. **a** $^{143}Nd/^{144}Nd$ versus Nb/La (weight ratio). These fluid immobile elements should be little influenced by fluid loss during subduction and so reflect protolith compositions. **b** Sr/Nd vs Ba/Th (weight ratios). These two ratios of fluid mobile over fluid immobile elements of similar magmatic incompatibility show the effects of subduction zone processes. Most of the meta-basalts have low Sr/Nd relative to mid-ocean ridge basalts (MORB), complementary to elevated Sr/Nd in arc lavas. Their Ba/Th are more variable, including some low values but ranging to higher values reflecting late phengite addition. Data sources: MORB is from compilation in ref. [70], ocean island basalts (OIB) and volcanic arc lavas compiled from PetDB (http://www.earthchem.org/petdb); Trace elements and Nd isotopes of samples from this study compiled from ref. [23,27,40] (see Supplementary Table 1), except for Nd isotopes of Cabo Ortegal Complex measured in this study

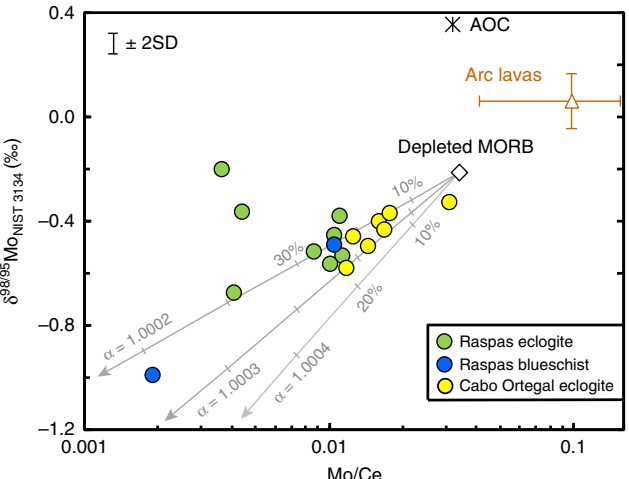

**Fig. 2** Observed and modelled variations in Mo isotope compositions as a function of Mo/Ce. Grey lines are model $\delta^{98/95}Mo_{NIST\ 3134}$ in residual oceanic crust following fractional loss of Mo-carrying fluids using different fractionation factors ($\alpha$) between fluid and eclogite at 2.6 GPa, 600 °C and an oxygen fugacity of FMQ + 4 (see Methods). Tick marks denote 10% increments of fluid loss by mass. We also plot reference values for: depleted mid-ocean ridge basalts (MORB) (ref. [21]), altered oceanic crust (AOC) using the Super Composite composition from ODP Site 801[17] and fluid-rich arc lavas[16-19,61], selected with Ba/Th > 400 to minimise complexity of variable sediment contributions. The representative sample uncertainty is from replicate analyses of W-2a.

$\delta^{98/95}Mo_{NIST\ 3134} = -0.31‰$, in keeping with its higher bulk $\delta^{98/95}Mo_{NIST\ 3134}$.

Our analysis of one eclogite that shows petrological signs of regression (SEC50-1), is not noticeably distinct from the others (Table 1), indicating the Mo systematics are robust to resetting and dominantly reflect prograde processes. Two serpentinites from the Raspas complex have high $\delta^{98/95}Mo_{NIST\ 3134}$ ( >0.1‰), presumably reflecting their interaction with isotopically heavy seawater ($\delta^{98/95}Mo_{NIST\ 3134} \sim 2.3‰$)[33] during serpentini-sation. Although the serpentinites are enriched in Mo relative to Ce (Table 1), their Mo abundances are still very low (<20 ng g$^{-1}$), and markedly lower than even the most Mo-depleted eclogites.

**High-pressure experiments**. In order to help account for our observations, we conducted experiments on the fractionation of Mo isotopes between rutile and hydrous siliceous melt. The latter is taken to represent subduction zone fluids, and though an imperfect analogue, it provides an experimentally convenient initial means to assess possible isotopic fractionation. It is important to note that Mo is dominantly tetrahedrally coordi-nated, as $Mo(VI)O_4^{2-}$ species, both in siliceous melts[34] and aqueous fluids[35]. In contrast, Mo occupies an octahedral site in rutile. Given co-ordination changes are a major driver of isotopic fractionation (e.g. ref. [36]), siliceous melt should therefore provide an adequate substitute for aqueous fluid for this preliminary investigation.

Experimental procedures are detailed in Methods and the results of the one successful experiment are summarised in Supplementary Table 2. We show that the liquid phase is isotopically heavier than the rutile ($\Delta^{98/95}Mo_{fluid-rutile} = 0.90 \pm 0.17‰$ at 600 °C, see Methods), consistent with the sense of fractionation predicted by rule of thumb changes in co-ordination of Mo between rutile and fluid/melt.

## Discussion

We argue that the marked depletion of Mo recorded by low Mo/Ce (Fig. 2) reflects its loss during extensive reactive fluid flow across the subducted crust[10,37], in keeping with low Sr/Nd in the eclogites (Fig. 1b). The coupled decrease in $\delta^{98/95}Mo_{NIST\ 3134}$ and Mo/Ce (Fig. 2) implicates isotopic fractionation is associated with Mo loss. The nearly two orders of magnitude variation in Mo/Ce indicates a variable degree of interaction with fluid as might be as anticipated from channelised flow during slab dehydration[10,38,39]. All samples have Mo/Ce lower than MORB and so Mo does not seem to be significantly affected by the subsequent process that adds some fluid mobile elements such as Ba (Fig. 1b). This addition reflects late fluid percolation that causes secondary growth of phengite. Such a fluid is unlikely to cause further rutile growth to host additional Mo but should be isotopically heavy (see below) such that re-equilibration will increase the $\delta^{98/95}Mo_{NIST\ 3134}$ of existing rutile, as we observe in one rutile sample (SEC43-1, Table 1), and move the bulk samples above the main array (Fig. 2) as is observed for two samples.

Pre-existing protolith Mo isotope heterogeneity[21] or variable interaction with sediment-derived melts[17,18] might be invoked to explain the Mo isotopic variability. However, the enriched MORB have higher rather than lower $\delta^{98/95}Mo_{NIST\ 3134}$ than depleted MORB[21] and although interaction with sediment-derived melts has been used to explain light Mo isotopes in some arc lavas[17,18], this scenario can be ruled out as the eclogites have MORB-like rare earth element patterns and Nd isotopic signatures[23,27,40].

As an initial quantification of the Mo systematics of the eclo-gites, we have modelled our data using a Rayleigh fractionation calculation (see Models). This simplistic approach provides a minimum estimate of the relative amount of fluid that has interacted with the residual solid. The slope of the data in Fig. 2 is essentially controlled by the fractionation factor ($\alpha_{fluid-eclogite}$), which we therefore empirically infer to be 1.0002–1.0004 (Fig. 2), corresponding to instantaneous fluid-rutile differences ($\Delta^{98/95}Mo_{fluid-rutile}$) of 0.9–1.1‰. These values are reassuringly similar in sign and magnitude to our experiments (0.9 ± 0.17‰, see above), despite the differences in fluid composition of our experiments and those experienced by the eclogites (see Methods for further discussion).

The magnitude of decrease in Mo/Ce and $\delta^{98/95}Mo_{NIST\ 3134}$ of a sample depends on its net Mo loss, which is a function of the fraction of fluid loss and Mo solubility in the fluid. Fortunately, the temperature and oxygen fugacity ($fO_2$) dependence of solu-bility of Mo in fluids at appropriate sub-arc conditions have been experimentally calibrated[15]. Since the temperature of rutile equilibration is independently monitored by its Zr concentra-tion[24] or other thermobarometry[22,23], our data allow us to examine the $fO_2$ of the fluids that transported Mo from the eclogites. Admittedly, we also need to stipulate residual miner-alogy and fluid salinity, but these are sufficiently well constrained for the sensitivity of the calculation to these parameters. We assume that the eclogites have garnet/clinopyroxene = 30:70 and contain 1.5% rutile[23], as set by the typical depleted MORB Ti contents of the samples. Evidence from experiments and natural rocks indicate that high-pressure fluids liberated by dehydration have low salinities[41] and this has been specifically documented for fluid inclusions trapped in the Raspas eclogites[30]. We use a fluid containing 5 wt% NaCl as an appropriate composition (para-meters and models are described in detail in the Methods).

We illustrate the isotopic effects of Mo removal by fluid loss from subducted crust in Fig. 3a. Significant Mo loss and asso-ciated isotopic fractionation in the residual eclogite is only achieved as a result of high Mo solubility at oxygen fugacities at least two orders of magnitude above the fayalite-magnetite-quartz buffer (FMQ + 2). Even for a fluid as oxidised as FMQ + 5, a

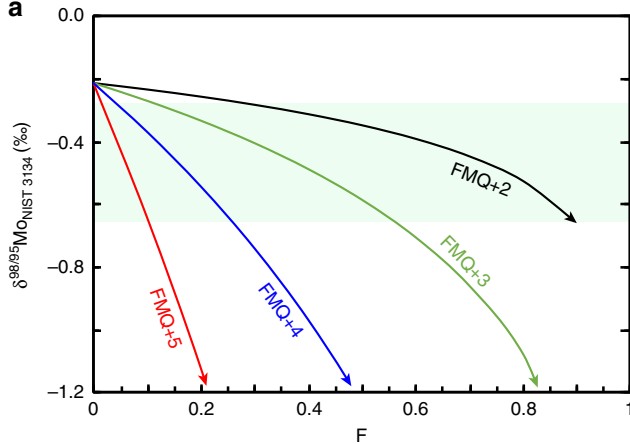

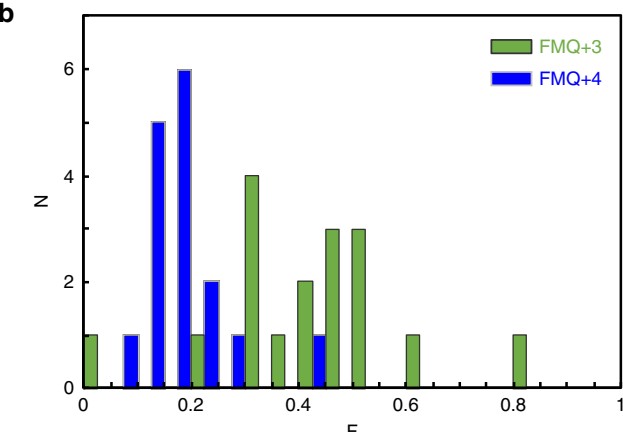

**Fig. 3** Modelled variations in Mo isotope composition as a function of mass fraction of fluid removed (F). **a** Results of the same model as in Fig. 2, but for different oxygen fugacities. Solid band represents the average $\delta^{98/95}Mo_{NIST\ 3134} \pm 1\,SD$ of eclogites and blueschists. **b** Histograms of modelled fractions of fluid removed, calculated for eclogites and blueschists (at two different oxygen fugacities) to illustrate the range of fluid loss required to account for their observed $\delta^{98/95}Mo_{NIST\ 3134}$

sizeable relative fluid loss (F > 0.05) is still required to reproduce the low $\delta^{98/95}Mo_{NIST\ 3134}$ we observe in our samples (Fig. 3a). This fraction of fluid loss is more than the upper bound of ~ 5% for the initial water content of subducting crust that can be estimated from analyses of the most altered, uppermost ~ 500 m of mafic oceanic crust[42] and much more than typical water contents < 0.6 wt% in unaltered, depleted MORB[43] which appears a more likely protolith of our eclogite samples (Fig. 2). Much larger fractions of fluid loss are required to model the isotopically lightest Mo compositions at lower $f$O₂ (Fig. 3a, b).

Not only is the amount of fluid that can be released from the subducted AOC limited but it is also unclear if such fluids will be sufficiently oxidising. There are few studies of the oxygen fugacity of prograde conditions recorded by exhumed eclogites that might have been imposed on fluids released, but the typical presence of pyrite inclusions in garnets[44] suggests modest $f$O₂, estimated to be a little below FMQ for Tian Shan eclogites[45]. Further evidence that an external source of fluid is key for perturbing the Mo systematics is that the most extreme $\delta^{98/95}Mo_{NIST\ 3134}$ and Mo/Ce are displayed by a blueschist, which must have experienced less loss of its initial water budget than an eclogite in a simple model of prograde fluid loss. Importantly, this low $\delta^{98/95}Mo_{NIST\ 3134}$ of the blueschist is not related to differences in protolith composition as Bezard et al.[21] showed that enriched

MORB (analogous to the Raspas blueschists, Fig. 1) have higher $\delta^{98/95}Mo_{NIST\ 3134}$ than depleted MORB (analogous to the Raspas eclogites).

An underlying serpentinised, slab mantle is an appealing potential source of oxidised and abundant fluids[10,30,46–50]. Serpentinites have low Mo abundances (Table 1) and so serpentine derived fluids will have the capacity to dissolve and transport Mo from the overlying crust, through which the fluids must pass as a consequence of subducted slab stratigraphy. Intrinsic, chemical heterogeneities within the serpentinites cause fluid flow channelization[12,38]. Although all eclogites will have experienced some fluid flow, not least to facilitate transformation to an eclogite-facies assemblage either by blueschist dehydration or gabbro-to-eclogite conversion[10], the variable Mo isotope systematics of the different samples (Fig. 2) indicates different fluid fluxes as would be anticipated from channelised flow.

Numerical models imply that uniform and pervasive porous flow is too slow to drain all water generated by dehydration of the slab, which requires high porosity, high permeability channels in a developing network[38]. Clear geological evidence for such major conduits is lacking. While there are good examples of smaller scale, vein features[37–39], they are not sufficiently numerous to have been the locus of major fluid flow. Additionally, such veins are more fracture-like structures and do not clearly document fluid interaction with a large enough volume to provide the major chemical fluxes to subduction zone magmatism. Meso-scale flow structures (i.e. at a dm to m or 'hand specimen' scale) with high permeability generated by, for example, porosity wave-like mechanisms[51,52] are thus required, but such permeability transients may leave no obvious record in the field[53]. We suggest that the Mo systematics provide a valuable tracer of such meso-scale channelisation. In order to remove and isotopically fractionate Mo, fluid flow has to be reactive and not fracture dominated. The variability of the Mo depletion from the rock volumes represented by our bulk samples further indicates a decimetre length-scale or more for such channel organisation.

The highly oxidising conditions required for our calculated fluid flow are striking, but in keeping with independent work. Calculations indicate that fluids released from serpentine can be between FMQ + 2 and FMQ + 5, depending on the buffering role of sulphides[48,54]. Matjuschkin et al.[55] have experimentally demonstrated that the powerful eight-electron transition from sulphate to sulphide is highly pressure dependant and easily shifted to these high $f$O₂ values above 2 GPa. Recent experiments and observations further point to an important role for oxidised, serpentine derived fluids in the subduction zone[54,56,57].

The isotopically light signature of exhumed depleted MORB eclogites makes an appropriate complement to isotopically heavy island arc lavas (Fig. 2). Although some sediment rich arc lavas also have elevated $\delta^{98/95}Mo_{NIST\ 3134}$[16,17] and some authors[16,19] have argued for a role of crystal fractionation in increasing $\delta^{98/95}Mo_{NIST\ 3134}$ (see additional discussion in ref. [58]), neither of these processes account for high $\delta^{98/95}Mo_{NIST\ 3134}$ in sediment poor, relatively mafic arc lavas[17,18]. The high Mo contents of arc lavas imply a substantial flux of isotopically heavy $\delta^{98/95}Mo_{NIST\ 3134}$. Freymuth et al.[17] argued that the high $\delta^{98/95}Mo_{NIST\ 3134}$ from ~500 m of AOC was insufficient to satisfy the Mo isotopic mass balance at the Mariana arc, requiring an additional source of isotopically heavy Mo. The Pb isotope systematics of the Mariana lavas further indicated that Pb, and by inference that of other fluid mobile elements, originated from largely unaltered, mafic crust[17]. Our study here illustrates a mechanism by which this source can supply isotopically heavy Mo (see Fig. 4).

Although we argue that serpentinites are the source of the fluid that results in the notable Mo isotopic fractionations, we stress

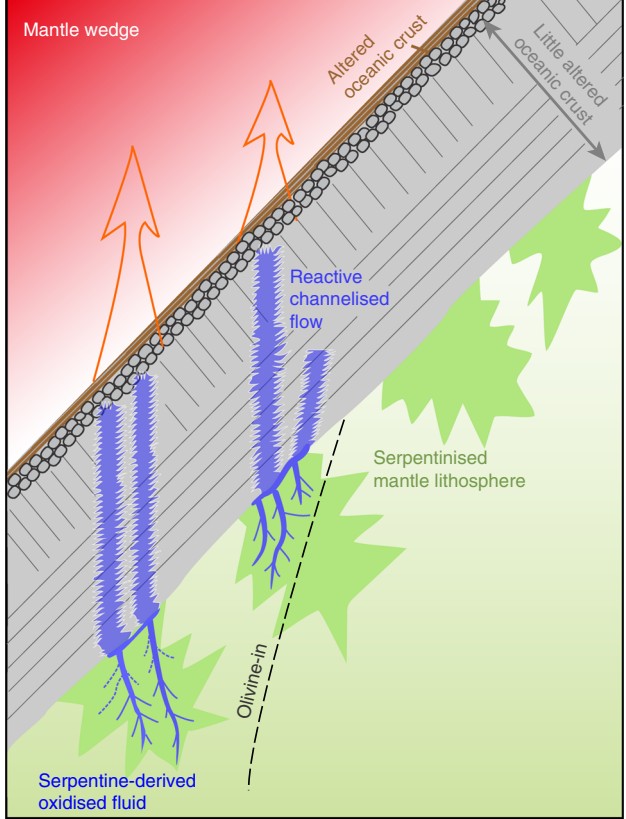

**Fig. 4** Cartoon of Mo isotope behaviour in subduction zones. The sketch indicates oxidised fluids from the dehydrating slab serpentine passing through the overlying oceanic crust, mobilizing Mo and leading to Mo isotopic fractionation. Resultant high $\delta^{98/95}Mo_{NIST\ 3134}$ fluids traverse into the hotter interior mantle wedge and induce melting. See text for details

that the intrinsic Mo budget of the serpentines themselves is not significant in the overall mass balance. The isotopically heavy nature of Mo in serpentines can modestly add to the high $\delta^{98/95}Mo_{NIST\ 3134}$ of fluids added to arc lavas, but the total amount of Mo in the serpentinites is an order of magnitude less than the amount of Mo lost by the fluid-fluxed eclogites. The serpentinised peridotite in the Raspas complex at peak metamorphic conditions (~2 GPa and ~600 °C) must have already lost water through brucite–antigorite breakdown (ref. [38,47]). Either this reaction, or alternatively terminal antigorite breakdown (>600 °C; ref. [47]), which occurred further down dip in the palaeo slab, provided the water to flux the eclogites.

An important consequence of our explanation of elemental and heavy isotope depletion of Mo in eclogites, and their complementary enrichment in arc lavas, is that oxidised fluids derived from serpentines must pass through the overlying crust via reactive flow. Our observations imply that crust and serpentinite are dominantly in their original stratigraphy, with the latter underlying the former (Fig. 4). This in turn argues against a melange zone as a dominant source of fluids for arc lavas (e.g., ref. [8]), which much less obviously requires passage of slab fluids through a coherent section of the oceanic crust.

In typical subduction zones, almost all isotopically light eclogites will be subducted to depth rather than exhumed. Thus, the distinctive Mo isotopic fingerprint imparted to the recycled crust should be evident in components of oceanic crust in ocean island basalts (OIB), if their sources are indeed dominated by recycled ocean crust. There are still limited Mo isotope analyses of OIB but $\delta^{98/95}Mo_{NIST\ 3134}$ lower than MORB values are observed by Liang

et al.[59]. Moreover, these authors reported a trend of decreasing Mo/Ce with lower $\delta^{98/95}Mo$ in a suite of OIB[59], consistent with the systematics that our study would predict. We further note that the over-print of fluid addition in some eclogite samples (Figs. 1b, 2) will add noise to this general relationship.

Our work illustrates the utility of the redox sensitive solubility of Mo in constraining fluid flow in the subduction zone. The extensive loss of Mo from subducted crust and its associated isotopic fractionation requires a large flux of oxidised fluid, most plausibly sourced from dehydration of underlying slab serpentine. Strong but variable perturbation of $\delta^{98/95}Mo_{NIST\ 3134}$ and Mo/Ce in all our samples of subducted crust suggests they experienced widespread, channelised, reactive fluid flow. As a result, these eclogites and blueschists carry the complementary signature to isotopically heavy and Mo enriched lavas observed in several island arcs.

## Methods

**Mo isotope measurements**. Mineral separates were prepared using a magnetic separator and hand picking. To remove surface contamination, mineral separates were washed for 10 min in methanol, cold 2 M $HNO_3$, and then rinsed with deionised water.

Mo isotope analyses of the samples were performed at the Bristol Isotope Group, School of Earth Sciences, University of Bristol, using a double spike technique, largely following the protocols described in ref. [60]. Unlike previous work undertaken in the lab, however, Parr bomb digestions were used to ensure complete dissolution of any rutiles which might be present in the eclogites. About 0.3–0.5 g powder was weighed into 25 ml PTFE beakers depending on the Mo concentration in each sample. For each 50 mg of sample, 1 mL reverse aqua regia ($HNO_3$:HCl = 3:1) and 0.5 mL HF were added. The beaker was then inserted and sealed in a high-pressure bomb before placing in an oven at 200 °C for 15–48 h. Sample solutions were then evaporated to incipient dryness and re-dissolved in 6 M HCl until a clear solution was obtained. The digested sample was finally spiked with an enriched $^{100}Mo$-$^{97}Mo$ double-isotope tracer and refluxed in 3 M HCl overnight in order to achieve sample-spike equilibration. We chose the geological reference material (GRM) W-2a to match the bulk-rock compositions and Mo concentrations of the eclogites, which we measured together with all batches of samples. We additionally measured GRM AGV-2 and JB-2, avoiding the heterogeneous, strongly Mo contaminated BHVO-2[60]. We made measurements of GRM using both high-pressure bomb and hotplate digestions, with no significant differences between results for either approach (Supplementary Table 3).

After chemical separation, samples were analysed using a multi-collector inductively coupled plasma mass-spectrometer (MC-ICPMS; Thermo Finnigan Neptune, serial no. 1020) aspirated as 50–70 ng g$^{-1}$ Mo solution in 0.4 M $HNO_3$−0.05 M HF. These Mo concentrations are lower than typically used in previous work from the laboratory and standards were run under the same conditions to provide accurate comparison. Our Ruthenium (Ru) doping experiments show that Ru interferences can be effectively corrected using measurements of either $^{99}Ru$ or $^{101}Ru$ (Supplementary Fig. 1). However, we observed occasional, non-systematic, large signals on mass 99. These signals were most likely due to polyatomic ions (e.g., $^{64}Zn^{35}Cl$, $^{40}Ar_2^{19}F$) rather than Ru itself given an absence of similar intensities for $^{101}Ru$, which we also monitored. Assuming such a signal on mass 99 was Ru would lead to an over-correction of the interference[17]. In order to avoid such artefacts, potential Ru interferences were corrected by both $^{99}Ru$ and $^{101}Ru$ for comparison, and data were rejected when the total 'Ru correction' was more than 0.1‰ in $\delta^{98/95}Mo_{NIST\ 3134}$. As previously observed in double spiked isotope analyses, average compositions of the reference standard within a measurement session often deviate slightly from zero. A correction was made for such drift by subtracting the average composition of NIST SRM3134 in a measurement session from each sample analysis in the same session.

The results are reported in standard $\delta^{98/95}Mo$ notation in per mil relative to NIST SRM3134, namely $\delta^{98/95}Mo_{NIST\ 3134} = [(^{98}Mo/^{95}Mo)_{sample}/(^{98}Mo/^{95}Mo)_{NIST\ SRM3134}] − 1]$. Replicate digestions and measurements of GRM of W-2a ($n = 13$) yielded $\delta^{98/95}Mo_{NIST\ 3134} = −0.03 \pm 0.04‰$ (2 SD), [Mo] = 0.41 ± 0.03 μg g$^{-1}$, JB-2 ($n = 5$) yielded $\delta^{98/95}Mo_{NIST\ 3134} = 0.03 \pm 0.04‰$ (2 SD), [Mo] = 0.97 ± 0.15 μg g$^{-1}$, and AGV-2 ($n = 5$) yielded $\delta^{98/95}Mo_{NIST\ 3134} = −0.16 \pm 0.04‰$ (2 SD), [Mo] = 1.88 ± 0.05 μg g$^{-1}$ (see Supplementary Table 3). The isotopic compositions and concentrations of Mo of these GRM agree within error with previously-published values[17,20,21,60,61]. The total procedural blanks ranged from 45 to 217 pg ($n = 6$), which are negligible for the samples.

**Nd isotope measurements**. Bulk-rock Nd isotope ratios of MORB-type eclogites from Cabo Ortegal Complex were measured using a Nu Plasma MC-ICPMS in the Institute of Oceanology, Chinese Academy of Sciences (IOCAS). About 50 mg of rock powders were dissolved with 1 ml reverse aqua regia ($HNO_3$:HCl = 3:1) and 0.5 ml HF in a high-pressure jacket equipped Teflon beaker at 190 °C for 15 h, which was then dried and re-dissolved with 2 ml 3 M $HNO_3$ for 2 h. The final

sample solution was first loaded onto Sr-spec resin to separate Sr, with the eluted sample solution collected and then loaded onto AG50W-X8 resin to separate rare earth elements (REE). The separated REE solution was dried and re-dissolved with 0.25 M HCl before being loaded onto Ln-spec resin to collect Nd. The measured $^{143}Nd/^{144}Nd$ ratios were normalised for instrumental mass fractionation using the exponential law to $^{146}Nd/^{144}Nd = 0.7219$. The international standard JNdi-1 was used as bracketing standard every four samples to monitor instrumental drift during the analysis. Repeated analysis for JNdi-1 gives an average $^{143}Nd/^{144}Nd = 0.512114 ± 0.000010$ ($n = 8$, 2 SD). The USGS rock standards of GSP-2 and BCR-2 run amongst the samples gave values of $0.511369 ± 0.000003$ (2SE) and $0.512629 ± 0.000003$ (2SE), respectively, consistent with the reported reference values ($0.511369 ± 0.000003$ and $0.512634 ± 0.000012$, respectively)[62].

**High-pressure experiments**. To explore the role of rutile in Mo isotope fractionation during subduction processes, we have performed a preliminary set of experiments. As crystallising rutile in an aqueous fluid is challenging, we have instead examined Mo isotope fractionation between rutile and wet andesitic melts. Starting compositions (see Supplementary Table 4) are based on the SKHDAN1 composition of ref. [63], doped with 467 µg g$^{-1}$ Mo added as MoO$_3$ powder. This was prepared by mixing dried oxide and carbonate powders with an agate pestle and mortar. Following decarbonation, the powder mixtures were homogenised by melting at 1400 °C for 12 h and then quenched and ground to a fine powder before this melting and quenching procedure was repeated.

Starting mixtures were loaded into Au$_{80}$Pd$_{20}$ capsules (3 mm outer diameter, ~7 mm length) after high purity water (182 kΩ m resistivity) was added to the capsule in an amount equivalent to ~5% by weight of the loaded starting mixture. Capsules were then sealed by welding. Loss of water was carefully monitored by comparing capsule weights before and after welding. Experiments were performed in a half-inch end-loaded piston cylinder at the University of Bristol using salt-pyrex assemblies with MgO fillers. Temperature was monitored with a C-type W-Re thermocouple. Experiments were performed at 1100–1200 °C and 2.0 GPa for 72 h prior to isobaric quenching by cutting the power.

Experimental evidence indicates that tetrahedrally coordinated Mo$^{6+}$ in silicate liquids is stable at oxygen fugacities higher than approximately one log unit below the iron-wüstite oxygen buffer[34,64–66]. This oxygen fugacity is several orders of magnitude lower than relevant for our study. The 'intrinsic fO$_2$' imposed by the assembly used in our experiments appears to be within one log unit of the Ni–NiO buffer (see ref. [67]), which is about 4 orders of magnitude above the iron-wüstite buffer. As this is sufficiently removed from oxygen fugacity induced changes in Mo speciation, fO$_2$ control was deemed not of primary concern here. We therefore applied no solid-state buffering system to control oxygen fugacity. Although Mo is generally a siderophile element, the high oxygen fugacity also limited loss of Mo to the Au$_{80}$Pd$_{20}$ capsule, as confirmed by a mass balance (see below).

The compositions of the quenched experiments were analysed by electron microprobe (Cameca SX-100) at the University of Bristol using synthetic and natural minerals and glasses as standards. Glass major element compositions were determined at 20 kV with a 4 nA beam and 10 µm spot size. These measurements were followed by analysis of Mo with a 1 µm diameter, 80 nA beam using three spectrometers to improve counting statistics. Subsequently, samples were crushed to a coarse powder (estimated grain size ~ 20 µm) and digested in pre-cleaned PFA beakers. Glass was preferentially dissolved by sonication in ~ 0.6 M HF at room temperature for ~1.5 h. After removing the supernatant, the remaining material was found to consist mostly of rutile grains (cf. Supplementary Fig. 2), which were digested in 28 M HF and a few drops of 15.5 M HNO$_3$ on a hotplate (150 °C). Although this method of rutile-glass separation is remarkably successful, our isotope fractionation factor likely represents a minimum due to minor amounts of remaining glass in the rutile fraction. We note that our approach relies on rutile being insoluble in 0.6 M HF and so quantitatively retaining its budget of Mo. In this respect the experiments of Wilkinson[68] provide a valuable reference. Wilkinson[68] documented negligible dissolution of rutile using more extreme conditions of ~15 M HF at 100 °C for 24 h.

Our experiments showed that the stability of rutile as single liquidus phase occurred in a very narrow temperature window (between ~1150 °C and its liquidus at ~1200 °C), with titanite appearing at lower temperatures. We present the results of a single experiment at 1175 °C in which titanite appeared in minimal abundance except for the outermost, colder ends of the capsule (Supplementary Fig. 3). The experiment contained ~14% rutile by volume and Mo contents of 360 ± 29 µg g$^{-1}$ and 472 ± 51 µg g$^{-1}$ in rutile and 4.4 wt% water-bearing andesitic melt, respectively (see Supplementary Table 2). From these Mo contents and phase abundances, we obtain a reconstructed bulk Mo content of 456 ± 44 µg g$^{-1}$. This concentration is indistinguishable from the Mo content of ~ 467 ± 67 µg g$^{-1}$ in the starting glass as determined by microprobe analysis, implying that even if any Mo loss occurred to the Au$_{80}$Pd$_{20}$ capsule the Mo systematics of the sample-capsule system were buffered by the Mo-rich sample.

The Mo isotope compositions of rutile and glass from this experiment were analysed on replicates that were obtained by splitting the coarse powder obtained from the central, titanite-free area of the capsule (Supplementary Fig. 3). The $\delta^{98/95}Mo_{NIST\ 3134}$ of the rutile replicates were $-0.28 ± 0.01$‰ and $-0.32 ± 0.01$‰ (2 SD) and those of the glasses $0.02 ± 0.05$‰ and $0.04 ± 0.01$‰ (2 SD) (see Supplementary Table 2). On average, rutile has $0.33 ± 0.06$‰ lower

$\delta^{98/95}Mo_{NIST\ 3134}$ compared with wet andesitic glass, corresponding to $\Delta^{98/95}Mo_{andesite-rutile} = 0.90 ± 0.17$‰ at an appropriate temperature for this study of 600 °C, after applying a temperature correction assuming no isotopic fractionation at infinite temperature and an inverse square relationship of the isotopic difference with absolute temperature.

**Model**. The Mo solubility in high-pressure (2.6 GPa) aqueous fluids is given by Bali et al.[15] as the following equation:

$$Log[Mo] = 0.44 × Log[fO_2] + 0.42 × [NaCl] - 1.8 × \frac{1000}{T} + 0.48 \qquad (1)$$

where concentrations are in molalities, fO$_2$ is oxygen fugacity and $T$ is temperature in Kelvin. Bali et al.[15] also presented Mo solubility in garnet, clinopyroxene and rutile at 2.6 GPa and 1000 °C. Together with the Mo solubility in fluid described above, fluid/mineral partitioning coefficients at 2.6 GPa and 1000 °C can therefore be determined, as presented by Bali et al.[15] in their Table 4. However, Bali et al.[15] did not investigate a temperature dependence for Mo solubility in the minerals, which hinders using Eq. (1) above to determine fluid/mineral partition coefficients at 600 °C, the temperature of interest in our study. As rutile is the dominant Mo host in the eclogites we studied, we have used the temperature dependence of Zr solubility in rutile[25] as analogue for the temperature dependence of Mo solubility in rutile to extrapolate the rutile solubility data from Bali et al.[15] at 1000 to 600 °C. The validity of this approximation is supported for our purposes by the observation that the temperature dependence of Zr-in-rutile is similar to the temperature dependence of Ti in quartz[69] and Ti in zircon[25]. The parameters used to calculate the fluid/mineral partition coefficients are given in Supplementary Table 5. The depleted MORB-type eclogite is assumed to have a mineral assemblage of garnet/clinopyroxene = 30:70 containing 1.5% of rutile (see the text). Applying these mineral abundance ratios and partition coefficients, the compositions remaining in the rocks ($c_s$) are calculated by Rayleigh distillation, given by the formula:

$$c_s = c_0 × (1 - F)^{\left(\frac{1}{D} - 1\right)} \qquad (2)$$

Where $c_s$ and $c_0$ are the concentrations (weight fractions) of Mo and Ce in the remaining rock (i.e., residual eclogite) and initial, subducted ocean crust (i.e., fresh MORB), respectively. $D$ is the bulk rock/fluid partition coefficient for the residual eclogite assemblages and $F$ is the proportion of fluid removed/flushed as mass fraction of the bulk rock. The mass fraction of Mo remaining in the rock is $f_s$:

$$f_s = (1 - F) × c_s/c_0 \qquad (3)$$

The $\delta^{98/95}Mo$ in the remaining rock is calculated using an approximate form of the Rayleigh distillation equation:

$$\delta^{98/95}Mo_s = \delta^{98/95}Mo_0 + (\alpha - 1) × \ln f_s \qquad (4)$$

where $\delta^{98/95}Mo_s$ and $\delta^{98/95}Mo_0$ are the $\delta^{98/95}Mo_{NIST\ 3134}$ of residual eclogite and initial subducted ocean crust, respectively, and $\alpha$ is the Mo isotope fractionation factor between fluid and bulk eclogite.

## Data availability
The authors declare that the data generated or analysed during this study are included in this published article and its Supplementary Information files.

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

## Acknowledgements

This study was funded by NE/M000419/1 and NE/L007428/1 building on work started in NE/J009024/1. S.C. was supported by Chinese NSF (Grant No. 41630968, 41130314, 41906050), Chinese NSF-Shandong (Grant No. U1606401), and the fellowship from the UCAS Joint Ph.D. Training Program (UCAS[2015]37).

## Author contributions

Analyses were made by S.C. and R.H. High-pressure experiments were undertaken by B.B., R.H. and R.B. T.E. designed the project. S.C. and T.E. wrote the paper. All authors contributed to discussions.

## Competing interests

The authors declare no competing interests.
