## [Peer Review File · Nature Communications]

Reviewers' comments:

Reviewer #1 (Remarks to the Author):

This paper presents stable Mo isotope data for eclogite samples that are being used as analogues for subducted oceanic crust (which was subsequently obducted and thus preserved on the Earth's crust).

The data is complemented with a set of experiments that is being used to underline the observations in the natural samples.

The main message of this paper is that dehydration of oceanic crust is not only constrained to the uppermost portion of the slab but extends towards the lower, serpentinitised sections. Based on the overall heavier Mo isotope signatures in the hypothesised fluids, the paper also aims to explain heavy Mo isotope signatures in arc lavas/continental crust.

In my opinion, concept no 1 – that of dehydration of serpentinites during subduction is nothing that is really groundbreaking or new. There are a range of recent studies in the last 10 years that argue for this. The geochemical signatures provided here *may be* used to support these findings (see below why maybe not), but doesn't add a fundamentally new concept or something that is surprising. Various authors have argued for the dehydration of serpentinites during subduction, including some authors of this study, e.g., John et al., EPSL, 2011, Debret & Sverensky SciRep 2017, Padron-Navara et al., 2011, JPET, Hattori & Guillot G3 2007 to name a few. So what is really new here that justifies publication in a Nature journal?

The second concept – that of explaining heavy arc lavas – is flawed in as much as arc lavas develop a heavy composition along a liquid line of descent. The lavas are not systematically biased towards heavier values as would be expected by an isotopically heavy fluid addition, but evolve towards heavier values. This demands fractionation being a key part in explaining the heavy signatures in arc lavas. See the compilation in Wille et al., 2018 Fig 2a. Looking at this, arc lavas clearly start at a mantle value. Arguing that each arc lava then represents a different degree of fluid addition, along with increasing silica is, in my opinion, deep in the realm of speculation and would need to be fleshed out.

An additional problem is the very heavy isotope signatures of boninites reported in Koenig et al – how do they fit in here? Koenig argues for the lightest signatures in the most slab fluid dominated samples. That contradicts the finding of this paper here. Are these uppermost slab fluids (AOC) and only deep slab fluids (serpentinites) are heavier? Maybe, but this would need to be discussed and elaborated in much more detail. Ignoring it does not add to a better understanding of Mo isotopes in arcs or processes of dehydration.

So overall I am not favor of publication in Nature Communication in its present form and some careful revisions are required. The paper makes a case and I am not denying that the authors interpretation of the data is in the realm of possibilities, but the case presented leaves many open questions and is very speculative.

Some more detailed comments

Paragraph lines 122f I found this paragraph somewhat confusing. The absence of Ti in "fluids", if correct, may be related to no rutile growth and thus no effect on Mo/Ce, but why would these fluids not carry any Ce, esp if "very" oxidised? Also – why would Ti not be mobile? As it is said elsewhere, the fluids here are at such high P (beyond 2nd critical endpoint) that these are melt-like and Ti would readily be transported. All of this is very speculative and qualitative.

The following discussion also sounds somewhat circumstantial. The authors admit that their experiments do not recreate deep crustal fluids (*imperfect analogue*). I appreciate that the authors do admit this hick up and it is also true that this may not matter too much as long as they are on the isotopically heavy or light side of the divide, but clearly this plays very much into mass balance, which is somehow required here.

I also struggle with primary and secondary fluids, some of which are argued to transport Mo (low Mo/Ce), others then also mobilize Ba (or not) but then don't affect Mo/Ce anymore? There is also much uncertainty in relation to the salinity, which is simply assumed but has a strong influence on Mo mobility and on the transport of other elements such as Ba.

Looking at Fig 2B – this is somehow in the world of speculation again. The FMQ+2 line is likely the most appropriate fO_2 for these conditions. Everything else is not really feasible. What on Earth would oxidise slab fluids above that ? The sulfide-sulfate transition is the key point – cant go past sulfate so what medium is bringing this up to FMQ + 5?!?! So accepting +2, this is a lot of Mo being transported, yet there is a lot of MO in rutile left behind. One would need some sort of basic mass balance to see how feasible this is.

Oliver Nebel, Melbourne Feb 2019

Reviewer #2 (Remarks to the Author):

In this study rutile is identified as the controlling mineral in retaining isotopically light Mo in the residual slab by a mechanism termed reactive flow: As serpentinite-derived slab fluids pass through the overlying layers of high-pressure rocks, they preferentially scavenge isotopically heavy Mo from rutile, thus explaining the isotopically heavy, complementary Mo signatures of arcs.

General

I find the the new Mo isotope data of highest quality and the study approach and the proposed model very interesting. albeit perhaps slightly specialized for a general readership of this journal. While it is new to suggest a complementary signature for Mo isotopes of some arc lavas based on actual data for rutile and eclogites/blueschists, it is not new that reactive fluid flow is an important mechanism in subduction zones. One example: Chen et al. write in Line 212: "... *further suggests channelised (but not fracture dominated), rather than uniform, pervasive flow.*"

A previous study already titles: "*Fluid escape from subduction zones controlled by channel-forming reactive porosity*", by Plümper et al. (2016) NatGeosc. Perhaps the authors could more stress on which mechanism regarding this process is newly constrained here or refined by Mo isotope results. I would also like to point out that, while I think that the below mentioned points need to be addressed before the sole involvement of pure fluids and not more complicated hydrous melts, I believe that the authors can do so and thereby improve the contribution to which I look forward.

Experiments

My concern is that hydrous melts are not fluids, which I note the authors admit may be an "*imperfect analogue*" (Line 102-103). What if the differences between these liquids result in different ways of leaching Mo from rutile? A number of previous studies pointed out that fluid-rich melts have higher mobilization potential compared to silica-poor fluids (Kessel et al., 2005 for a prominent reference). This aspect may be more critical than expected as some samples do not quite fit in overall trends (see below for actual examples). It would help if the authors addressed some of these apparent discrepancies and perhaps found that the direction of fractionation (if not the extent) by melts and fluids remains in order to come to a qualitative rather than quantitative conclusion. This strategy should be even more followed in terms of calculating actual FMQ.

In line 370 it says: "*Glass was preferentially dissolved by sonication in ~0.6 M HF at room temperature for ~1.5 hours. After removing the supernatant, the remaining material was found to consist mostly of rutile grains (cf. Supplementary Fig. 2), which were digested in 28 M HF and a few drops of 15.5 M HNO₃ on a hotplate (150°C). Although this method of rutile-glass separation is remarkably successful, our isotope fractionation factor likely represents a minimum due to minor amounts of remaining glass in the rutile fraction.*" Is there a reference for this method or can the authors explain why the first preferential dissolution of glass does not cause collateral-side effects like even minor leaching of isotopically heavy Mo from rutile?

Line 391: "*corresponding to $\Delta_{98/95}\text{Mo}_{\text{andesite-rutile}} = 0.90 \pm 0.17 \text{ ‰}$ at an appropriate 392 temperature for this study of 600°.*"

Top-slab temperatures at sub-arc depths are likely to be 700–900°C (Hermann and Spandler, 2008). Also, sediment melts seem almost always required to explain element transport across subduction zones. In the light of evidence for metasediments in the

Raspas complex (Line 64), I think perhaps the authors need to better discuss the role of sediments melts in the light of their melt-involved experiments (see also Point 4 under Discussion below).

Presentation

It needs to be highlighted what an excellent documentation of analytical method and data quality regarding both the Mo isotope- and experimental work is presented here. The latter is comendable and the isotope group shows how it should be done.

One significant problem in this manuscript (+ supplement) is that insufficient geochemical background data for the samples are provided. This is in very stark contrast to the comendable point above.

Given the scope of the journal and the implications the authors suggest, neither a reviewer nor a reader should have to look up additional references for Ce (marked with citation in the table) to croscheck or build different diagrams. Moreover, Ba, Th, Nb, La etc... are not even cited although shown in diagrams. It is also important to know if for some samples such data (where not in diagrams) are unavaiable or only for different sample splits etc. The latter would not be generally bad but nevertheless important to know. For instance Table 1 contains 10 eclogite/blueshist data and Fig. 1b only 8., so missing two which according to supporting literature “*Subducted seamounts in an eclogite-facies ophiolite sequence: the Andean Raspas Complex, SW Ecuador*” should be there (see Fig R1).

Currently such things are not transparent. This undermines the otherwise excellent work. I do not think that such practice should be accepted.

Fig. R1 All data from the Raspa Complex where Sr, Nd, Ba and Th are provided according to supporting literature “*Subducted seamounts in an eclogite-facies ophiolite sequence: the Andean Raspas Complex, SW Ecuador*”.

Besides being a matter of comprehensiveness, there are also scientific considerations: Rutiles from only one sample were analyzed and although the results seem convincing to support the main conclusion, I find it important to balance these limitations and have more comprehensive background geochemical data for the bulk samples to place them in the right context/to see if they are purely affected by fluid involvement (and not more complex hydrous melts). Some specific examples are given below.

Discussion

As much as I tend to agree with the model that rutile may be the complementary, light Mo-bearing mineral host compared to slab fluid-enriched mantle wedges, I would like to see the bulk samples better scrutinized.

- 1) What do bulk trace element systematics contribute to the issue of seawater contamination? In Line 95 it says that, “*Two serpentinites from the Raspas complex have high $\delta^{98/95}\text{Mo}_{\text{NIST 3134}}$ ($>0.1\%$), presumably reflecting their interaction with isotopically heavy seawater ($\delta^{98/95}\text{Mo}_{\text{NIST 3134}} \sim 2.3\%$) during serpentinitisation. Although the serpentinites are enriched in Mo relative to Ce (Table 1), their Mo abundances are still very low (<20 ng/g) and markedly lower than even the most Mo-depleted eclogites.*”
First, why then are these samples included in the eclogite suite, according to the Figure 1a legend, if they are serpentinites?
In order to follow the authors explanation, where these two seawater contaminated samples plot in Fig. 1a (to see if slab fluid loss has clearly occurred according to Ba/Th) and why they show lower Mo/Ce (clearly a slab fluid-loss signature which leaves Mo heavy?).
- 2) Further, starting in line 115 it says: “*The latter observation implies that Ba is mobilised by fluids and removed from some samples, but local fluid addition can have an opposite effect*”. If this is true should the repeated fluid removal and re-infiltration processes produce a trend in Ba/Th vs Mo isotopic composition? If it is less crucial whether blueschist or eclogite was present rather than the amount of fluid-rutile interaction and hence extent of reactive flow occurred, then the proxy for fluid scavenging or re-infiltration could show trends with the magnitude of fractionation. However, it seems from Fig 1b that most of the Raspas samples have seen more fluids than the other samples investigated here. Yet these show the lowest Mo/Ce and Mo isotopic composition. Would not the opposite be expected from the authors statement above (Line 115)? Fluids that pass through the slab scavenge some Mo from rutile and are gone, thereby leaving less Mo where lower amounts of fluids remain? Why then are the samples with the lowest Mo/Ce the ones with the highest Ba/Th? This is at least counter intuitive and needs an explanation. I would therefore also be careful with accepting the statements in lines 116 to 121, which a reader cannot follow without the relevant trace element data.
- 3) It is not properly discussed what the consequences are for that blueschist samples do not just show a typical N-MORB signature. In Line 67 it says: “*Most Raspas eclogites have geochemical characteristics indicative of a mid-ocean ridge basalt (MORB) protolith, whereas associated blueschists exhibit more enriched, seamount-like characteristics.*” Then in Line 212: “*There are still limited Mo isotope analyses of ocean island basalts (OIB) but $\delta^{98/95}\text{Mo}_{\text{NIST 3134}}$ lower than mid-ocean ridge basalt values are observed by Liang et al.42.*”
I note one red sample (notably a blueschist) that shows the lightest Mo isotopic composition. According to statement from line 212 this could then be a previously generated signature? Is this a chicken and egg issue? Well this sample certainly shows the highest Nb/La (at least I infer this but again do not have the actual data in a table). If so, then it is in principle ok to discuss it as affected by slab fluids (even twice...). However the quantification of the singular process (or relevance of Mo isotope value here) should be done without the blueschists. In any case this issue requires attention.
- 4) It remains unclear why starting line 255 (Fig.2 legend) it says “arc lavas were selected with Ba/Th >400 to minimise complexity of variable sediment contributions”,

when in Line 64 it says: “The Raspas Complex, Ecuador, contains metasediments...” Is this not contradictory? Why exactly is the dataset trimmed? What role do subducted sediments play in the system? Evidence exists for that hydrous melts are needed for efficient extraction of LILEs from subducted sediments (Hermann and Spandler, 2008). Can it be ruled out that sediment melts are involved in the former subduction setting and match hydrous melt-derived experimental results? Is the conclusion that merely fluids cause the observed signatures valid? Please keep in mind apparent discrepancies with Ba/Th pointed out above.

Further

Line 56

what are typical oceanic sediments? Oceanic sediments display a large range of Mo isotopic composition from heavy signatures in reducing sediments, up to -2.1 permil down to -0.8 permil in deep sea pelagic sediments.

Line 106

Is this a combined effect of Mo(IV) bound octahedrally in rutile and MoO₄²⁻ as Mo (VI) in silica melt? Please specify!

Line 363: *“No buffering system was applied to control oxygen fugacity, but previous experience with our piston cylinder has shown that the oxygen fugacity in such experiments buffers around Ni-NiO.”*

This requires a bit more explanation and at least a reference but also an idea about the uncertainty involved

Reviewer #3 (Remarks to the Author):

Subduction zones, plate margins where one tectonic plate is forced below the other and thus dragged into the Earth's mantle, are unique geodynamic features of Earth that keep it a living - and thus habitable - planet. Through the release of fluids (among which CO₂) back to the surface, subduction zones regulate the Earth's climate over geological timescales. The transport of cold crustal material into the hot Earth's mantle also plays an important role in maintaining planetary-scale mantle convection cells by which heat from the core-mantle boundary is transported to the Earth's surface. A clear understanding of the geological processes that operate in subductions zones is therefore vital to research fields far beyond those of Earth Sciences.

In this respect, this contribution adds a key element of information to this topical issue. It has long been thought that fluids released into the overlying arc volcanoes (and ultimately into the atmosphere) originate from the upper part of subducting plate. This part has been in contact with seawater during the million-years-long journey of tectonic plates from mid-ocean ridges, where they are formed by partial melting of the Earth's mantle, into the subduction zones, where they are recycled into the mantle as subducting slabs. Yet, although dehydration of fluids contained in the upper parts of subducting slabs may contribute to the volatile budget of arcs, its sheer volume appears to be too small.

Using state-of-the-art Mo isotope systematics, Chen and co-workers were able to draw a completely different picture of how slab dehydration may operate. Rather than a superficial, small-volume origin of fluids, their data lend important support for a model, whereby high-volume, water-rich mantle material underlying the cold (and dry) oceanic crust, releases its fluid-content at mantle depth, thus flushing the overlying (largely dry) slab with large amounts of supercritical fluids. Not only may this help to reconcile the volumes of fluids released into the volcanic arc but has also important implications for the chemical budget of the fluids that they are able to obtain during their passage through the chemically enriched subducting slab.

On top of that, the data also links up two important recent observations in isotope geochemistry: Volcanic rocks from subduction zones have an (overall) "heavy" Mo isotope composition, whereas mantle-derived rocks that potentially contain a recycled component have a "light" Mo isotope composition. The experiments and measurements by Chen and co-workers, for the first time, support a recently suggested hypothesis, that the dehydration and recycling process of tectonic plates in subduction zone is the driving force that caused these somewhat complimentary isotopic features.

The study itself has been meticulously designed and carried out. Both, experiments and isotope data adhere to the highest quality levels and there is little room for improvement in the overall assessment and interpretation of the data.

I have one minor comment that does not really affect the overall outcome of the study. In turn, I think that addressing it in a mildly revised version may help to strengthen the authors' case: The experiments have been carried out in Au-Pd capsules. Given that Mo is moderately siderophile and may thus form alloys with the capsules, it would be good to address this issue with respect to the partitioning (and thus fractionation) behaviour of Mo between the silicate melt and rutile. On the one hand, I reckon, the system will be heavily buffered by the high Mo concentration of the glass.

On the other hand, only the relative isotopic difference between rutile and melt is of interest, not their absolute dMo values. Maybe this would be worth mentioning in the method section.

There are a few minor quibbles that are addressed below. Overall, I regard this manuscript as an excellent and novel contribution of potential interest to researchers beyond the field of Earth Sciences. It is well worth being published in Nature Communications following some (very) minor revisions.

Detailed comments:

Line 62: I reckon adding 'subducted' to oceanic lithosphere would emphasize the complementarity mentioned.

Lines 69 and 72: Second half of sentence sounds a bit odd. Maybe rephrase?

Line 109ff: This sentence is a bit confusing. How about: "We argue that the depletion of Mo in the eclogites compared to MORB, as recorded by their lower Mo/Ce, reflects the loss of Mo from the subducting crust to fluids that traverse the dehydrating slab. This is in keeping ...".

Line 178: Wouldn't 'take up' be a better expression than 'dissolve'?

Lines 186 and 187: Please clarify the nature of these experiments and observations. Isotopic, chemical, etc ...

Lines 205 to 207: The statement "further down-dip of the paleo slab" seems to contradict the pressure estimates of the two breakdown reactions given (both at 2GPa).

Line 288 (and below): Please change from 'rock standard' to 'geological reference material' (GRM) as neither W-2a nor any other reference material issued by the USGS or GSJ are standards.

Line 404: 'Mo solubility in the minerals' sounds odd. Do you mean Mo partitioning into (or between) minerals?

Reviewer 1#

This paper presents stable Mo isotope data for eclogite samples that are being used as analogues for subducted oceanic crust (which was subsequently obducted and thus preserved on the Earth's crust).

The data is complemented with a set of experiments that is being used to underline the observations in the natural samples.

The main message of this paper is that dehydration of oceanic crust is not only constrained to the uppermost portion of the slab but extends towards the lower, serpentinitised sections. Based on the overall heavier Mo isotope signatures in the hypothesised fluids, the paper also aims to explain heavy Mo isotope signatures in arc lavas/continental crust.

[1] In my opinion, concept no 1 – that of dehydration of serpentinites during subduction is nothing that is really groundbreaking or new. There are a range of recent studies in the last 10 years that argue for this. The geochemical signatures provided here may be used to support these findings (see below why maybe not), but doesn't add a fundamentally new concept or something that is surprising. Various authors have argued for the dehydration of serpentinites during subduction, including some authors of this study, e.g., John et al., EPSL, 2011, Debret & Svervensky SciRep 2017, Padron-Navara et al., 2011, JPET, Hattori & Guillot G3 2007 to name a few. So what is really new here that justifies publication in a Nature journal?

We thank the reviewer for his frank comments which emphasise points which we need to make with greater clarity. This has been very helpful.

Response 1#:

We evidently did not make clear what we see as the full novelty of our findings. In revision we have tried to address this in several respects.

1. We agree that the role of serpentine dehydration in supplying water to subduction zones has become quite widely (although by no means universally) accepted in recent years. Yet there are two quite different models for the role of serpentine in the subduction zone, usefully summarized in Spandler and Pirard (2013). In one the serpentine forms above the down-going slab, in the other slab mantle is serpentinitised. The former is associated with a subduction mélange model that has gained recent traction in linking geochemical and field observations, whereas the latter offers a means to flux elements from the largely unaltered, downgoing crust into the sub-arc. Both models are widely invoked. Distinguishing between these scenarios has major ramifications for the operation of the subduction system. Our data make a strong case for the latter, with attendant implications for elemental fluxes carried from the oceanic crust. We have significantly expanded our discussion of this debate in the Introduction.

2. Our work illustrates a powerful geochemical means to trace the fluid flux experienced by exhumed eclogites. The integrated record of reactive fluid flow preserved in the Mo systematics of eclogites demonstrates spatial variability that indicates channelization on a bulk sample scale, as anticipated for the major fluid flux expected from serpentine dehydration but previously not observable. While there have been number of fine scale

petrographic studies of veins, these cannot mark the routes of major fluid flow. While the petrology of the major channelized network seems largely overprinted, the geochemical record is still present in the Mo isotope signatures. We further emphasise this novel contribution of our work in the revised manuscript.

3. We show a striking complementary relation between elemental depletion and isotopic fractionation in the residual slab with elemental enrichment and isotopic fractionation in an opposite sense evident in literature work on arc lavas. Demonstrating this relationship is a first. Given the major flux of elements carried in arc magmatism this linkage makes clear the significance of the hand-specimen observations we make in this study.

[2] The second concept – that of explaining heavy arc lavas – is flawed in as much as arc lavas develop a heavy composition along a liquid line of descent. The lavas are not systematically biased towards heavier values as would be expected by an isotopically heavy fluid addition, but evolve towards heavier values. This demands fractionation being a key part in explaining the heavy signatures in arc lavas. See the compilation in Wille et al., 2018 Fig 2a. Looking at this, arc lavas clearly start at a mantle value. Arguing that each arc lava then represents a different degree of fluid addition, along with increasing silica is, in my opinion, deep in the realm of speculation and would need to be fleshed out.

Response 2#: It was an oversight not to flag this debate in the original manuscript and in the Introduction of the revision we rectify this omission. Further, in the revised Discussion we point to the arguments against a differentiation-driven mechanism of high $\delta^{98/95}\text{Mo}$ in the mafic samples considered in this manuscript.

While important that we acknowledge the case made for fractionation of Mo isotopes during arc-lava differentiation, we note that at most this process influences some evolved lavas that are not the focus of discussion here. Willbold and Elliott (2017) presented detailed arguments against a major role for Mo isotope fractionation during arc lava differentiation. We cite this study in the revision and reprise these points below, together with comments on the further issues raised by the reviewer and Wille et al. 2018.

The reviewer's suggestion that magmatic differentiation could potentially fractionate Mo isotopes was first proposed by Voegelin et al. (2014) who reported an increase in $\delta^{98/95}\text{Mo}$ from basalt to rhyolite in a suite of Kos Plateau Tuff samples. Yet Willbold and Elliott (2017) noted that the majority of the Kos samples have a remarkably uniform Mo isotope composition ($\delta^{98/95}\text{Mo} = 0.23 \pm 0.03\text{‰}$; 2SD, Fig. R1). Only two samples have consistently lower $\delta^{98/95}\text{Mo}$ of 0.08‰ ($\pm 0.05\text{‰}$; 2SD). This results in an almost step-like increase in $\delta^{98/95}\text{Mo}$ within a very narrow range of Mo concentration. Such a sharp change is hard to explain with fractionation crystallisation of hornblende and biotite alone (as proposed by Voegelin et al, 2014), since this transition occurs at the mafic end of the suite before these hydrous phases would be expected to be on the liquidus. Thus, Willbold and Elliott (2017) concluded that differentiation appears to have a limited effect on Mo isotope data in magmatic rocks.

Whilst Wille et al. (2018) argued for Mo isotopic fractionation during differentiation, this is not borne out by the data (see Fig. R1 below). A plot of SiO_2 (or MgO , not shown) vs $\delta^{98/95}\text{Mo}$ (Fig. R1) shows no significant correlation, either for the Banda arc samples or the

global dataset. Wille et al. (2018) excluded the isotopically lighter Banda arc samples (which they argued has suffered assimilation), but even doing this a trend is scarcely evident in the remaining samples (Fig. R1). Regardless of whether or not some Mo isotopic fractionation does occur during differentiation, the compilation of arc lavas we discuss consists of mafic arc lavas with Ba/Th > 400. This data set was chosen specifically to avoid complexities arising by potential Mo isotope fractionation during differentiation (i.e. the mafic criterion) or potential involvement of sediment melts (i.e. Ba/Th criterion). These samples should thus be unaffected by any of the processes inferred from the sample sets of Voegelin et al. (2014) or Wille et al. (2018). Moreover, it is notable that the most mafic samples in both of these studies themselves have $\delta^{98/95}\text{Mo}$ higher than the typical mantle value of -0.25‰ (Fig. R1). In all, we argue that evidence for Mo isotopic fractionation during arc lava differentiation is not well supported and even if it does occur in the most evolved lavas it does not influence our discussion of more mafic arc lavas.

Figure R1: Plot of SiO_2 vs $\delta^{98/95}\text{Mo}$ shows no significant correlation, either for a) the Banda arc samples (samples excluded by Wille et al. 2018 as open symbols) or b) the global dataset (Aegean Arc are the Kos Plateau Tuff samples of Voegelin et al. 2014).

[3] An additional problem is the very heavy isotope signatures of boninites reported in Koenig et al – how do they fit in here? Koenig argues for the lightest signatures in the most slab fluid dominated samples. That contradicts the finding of this paper here. Are these uppermost slab fluids (AOC) and only deep slab fluids (serpentinites) are heavier? Maybe, but this would need to be discussed and elaborated in much more detail. Ignoring it does not add to a better understanding of Mo isotopes in arcs or processes of dehydration.

Response 3#: It seems that the reviewer has mis-remembered the paper by König et al. (2016). They find the heaviest Mo isotopic compositions are observed for the most slab fluid dominated samples and isotopically lighter signatures are related to increasing relevance of terrigenous sediment subduction and sediment melt components. Thus, they conclude that their observation complements previous conclusions of Freymuth et al. (2015) that an isotopically heavy Mo fluid flux likely results from selective incorporation of isotopically light Mo into secondary minerals within the subducting slab. This is exactly the model we propose in our manuscript.

So overall I am not favor of publication in Nature Communication in its present form and some careful revisions are required. The paper makes a case and I am not denying that the authors interpretation of the data is in the realm of possibilities, but the case presented leaves many open questions and is very speculative.

We hope our revisions and responses now make the case more persuasively.

Some more detailed comments

[4] Paragraph lines 122f I found this paragraph somewhat confusing. The absence of Ti in “fluids”, if correct, may be related to no rutile growth and thus no effect on Mo/Ce, but why would these fluids not carry any Ce, esp if “very” oxidised? Also – why would Ti not be mobile? As it is said elsewhere, the fluids here are at such high P (beyond 2nd critical endpoint) that these are melt-like and Ti would readily be transported. All of this is very speculative and qualitative.

Response 4#: We agree that HFSE and REE can be transported in high temperature fluids, however, John et al. (2010) showed empirically that the HSFE and REE are fluid immobile trace elements in the studied eclogites, which show depleted, MORB-like patterns. For Ti and the REE to be fluid immobile effectively requires residual rutile and zoisite in the fluid percolation pathway (as is widespread in blueschist facies likely in the deeper sections of the crust). By contrast, the fluid-mobile LILE scatter widely with up to ten times enrichment or depletion in the eclogite samples, which indicates the combined effects of general fluid loss sometimes superimposed by late fluid addition (see our Fig. 1b).

This is an important point and we appreciate that we did not make this sufficiently clear. We have changed the text to introduce the overall concept at an early stage and then apply it specifically to Mo (and Ce) later.

[5] The following discussion also sounds somewhat circumstantial. The authors admit that their experiments do not recreate deep crustal fluids (imperfect analogue). I appreciate that the authors do admit this hick up and it is also true that this may not matter too much as long as they are on the isotopically heavy or light side of the divide, but clearly this plays very much into mass balance, which is somehow required here.

Response 5#: We thought it important to acknowledge that the fluid we used in the experiments was an imperfect analogue, but retrospectively this gave misleading over emphasis. In revision, we have highlighted two major considerations about the suitability of the system we used:

1. What drives many isotopic fractionations is a change in co-ordination. For our case, there is a contrast between an octahedral host lattice site for Mo in rutile and a tetrahedral co-ordination in either aqueous fluid or silicate melt.
2. The numerical value we use for the fractionation factor is determined empirically by the slope of the array in Fig 2. As we show in the supplementary material and flag in the text, this slope is essentially controlled by the fractionation factor. Hence our experimental measurement is a valuable consistency check, but the fractionation factor used in modelling is pinned by the observations. We note that the mobility of Mo, to which the modelling is sensitive, is parameterized from experiments using experiments with appropriate aqueous

fluids (Bali et al., 2012). It is only the experimental confirmation of the fractionation factor which we have undertaken using a siliceous melt.

Finally, we should note that despite a burgeoning literature of ‘non-traditional’ isotopic measurements, the field is under-pinned by very few experimental determinations of fractionations. Our coupling of experimental and observational data is rare.

[6] I also struggle with primary and secondary fluids, some of which are argued to transport Mo (low Mo/Ce), others then also mobilize Ba (or not) but then don’t affect Mo/Ce anymore? There is also much uncertainty in relation to the salinity, which is simply assumed but has a strong influence on Mo mobility and on the transport of other elements such as Ba.

Response 6#: We now spend more time in the early part of the manuscript reviewing the existing evidence for multiple phases of fluid flow affecting the eclogites (John et al., 2010). In the new discussion we also further discuss localization of fluid pathways in space and time (e.g., Bloch et al., 2018; John et al., 2012; Plümper et al., 2017; Taetz et al., 2018). This clarifies that a specific rock volume might be affected by multiple pulses of externally-derived fluids. We cite evidence for the low salinity of slab derived fluids that interacted with eclogites in general and Raspas in particular. This is relatively well constrained.

[7] Looking at Fig 2B – this is somehow in the world of speculation again. The FMQ+2 line is likely the most appropriate fO_2 for these conditions. Everything else is not really feasible. What on Earth would oxidise slab fluids above that? The sulfide-sulfate transition is the key point – can’t go past sulfate so what medium is bringing this up to FMQ + 5?!?! So accepting +2, this is a lot of Mo being transported, yet there is a lot of MO in rutile left behind. One would need some sort of basic mass balance to see how feasible this is.

Response 7#:

In our work, we do not claim that we constrain the oxygen fugacity to any one of the values we show in Fig 2b,c (now Fig 3a,b). We merely state that below FMQ+2 Mo is nearly immobile in fluids, so the oxygen fugacity needs to be $FMQ \geq +2$. The range of oxygen fugacities above FMQ+2 that we use are not chosen arbitrarily but taken from the work of Debret and Sverjensky (2017), which uses petrology and thermodynamics to model oxygen fugacities of fluids liberated during serpentinite dehydration.

In terms of the mass balance of Mo transported and left behind in rutile, this is implicit in the modelling of isotopic fractionation and change in Mo/Ce.

We have tried to highlight these points in the main text as well as in the modelling methods.

Oliver Nebel, Melbourne Feb 2019

Reviewer2#

Review of MS NCOMMS-18-37350 by Chen et al.

In this study rutile is identified as the controlling mineral in retaining isotopically light Mo in the residual slab by a mechanism termed reactive flow: As serpentinite-derived slab fluids pass through the overlying layers of high-pressure rocks, they preferentially scavenge

isotopically heavy Mo from rutile, thus explaining the isotopically heavy, complementary Mo signatures of arcs.

General

[8] I find the new Mo isotope data of highest quality and the study approach and the proposed model very interesting, albeit perhaps slightly specialized for a general readership of this journal. While it is new to suggest a complementary signature for Mo isotopes of some arc lavas based on actual data for rutile and eclogites/blueschists, it is not new that reactive fluid flow is an important mechanism in subduction zones. One example: Chen et al. write in Line 212: "... further suggests channelised (but not fracture dominated), rather than uniform, pervasive flow." A previous study already titles: "*Fluid escape from subduction zones controlled by channelforming reactive porosity*", by Plümper et al. (2016) NatGeosc. Perhaps the authors could more stress on which mechanism regarding this process is newly constrained here or refined by Mo isotope results.

Response 8#:

We can see that we did not make clear the novelty of our observations for constraining fluid flow in the subducted slab. We hope we have rectified this in revision. In brief, while Plümper et al. (2017) (and a number of other studies) document small scale features of channelization a record of larger (meso-)scale permeability required to drain the dehydrating slab has not been evident. It is likely that original petrographic features are over-printed and so the chemical variability of Mo loss at the hand-specimen scale represents the first direct evidence of these processes.

Please also see Response 1#.

[9] I would also like to point out that, while I think that the below mentioned points need to be addressed before the sole involvement of pure fluids and not more complicated hydrous melts, I believe that the authors can do so and thereby improve the contribution to which I look forward.

Experiments

[10] My concern is that hydrous melts are not fluids, which I note the authors admit may be an "*imperfect analogue*" (Line 102-103). What if the differences between these liquids result in different ways of leaching Mo from rutile? A number of previous studies pointed out that fluid-rich melts have higher mobilization potential compared to silica-poor fluids (Kessel et al., 2005 for a prominent reference). This aspect may be more critical than expected as some samples do not quite fit in overall trends (see below for actual examples). It would help if the authors addressed some of these apparent discrepancies and perhaps found that the direction of fractionation (if not the extent) by melts and fluids remains in order to come to a qualitative rather than quantitative conclusion. This strategy should be even more followed in terms of calculating actual FMQ.

Response 10#:

The P-T conditions of our samples are ~ 600°C and ~2GPa, well below the transitional fluids investigated by Kessel et al. (2005). Aqueous fluids at these conditions have the potential to mobilize elements (see Beinlich et al., 2010; Gao et al., 2007; Herms et al., 2012; John et al., 2004) but they are not yet transitional fluids (e.g., Hermann et al., 2006). Herms et al. (2012)

specifically characterised the fluid inclusions and the elements they mobilise for the Raspas complex eclogites used in this study. We mention this point explicitly in revision. In terms of modelling the behavior of Mo, we use the only existing solubility data (Bali et al., 2012) but these detailed experiments were conducted at highly appropriate conditions for our samples and provide us with unusually good constraints on this process. We have further highlighted this in the revision.

Please see Response 5#.

[11] In line 370 it says: *"Glass was preferentially dissolved by sonication in ~0.6 M HF at room temperature for ~1.5 hours. After removing the supernatant, the remaining material was found to consist mostly of rutile grains (cf. Supplementary Fig. 2), which were digested in 28 M HF and a few drops of 15.5 M HNO₃ on a hotplate (150°C). Although this method of rutile glass separation is remarkably successful, our isotope fractionation factor likely represents a minimum due to minor amounts of remaining glass in the rutile fraction."*

Is there a reference for this method or can the authors explain why the first preferential dissolution of glass does not cause collateral-side effects like even minor leaching of isotopically heavy Mo from rutile?

Response 11#: This is an important point that we should have discussed in the original version. We have now reported the evidence that gave us confidence in our approach in the supplementary information and below.

The PhD dissertation by Wilkinson (2015), tries both low pressure and temperature (LPT) and high-pressure and temperature (HPTS) methods to dissolve eclogite powders. In the LPT methods, 0.095 - 0.125 g of sample was weighed into a sealable PTFE (Teflon™) beaker and approximately 10 ml of 28M hydrofluoric acid (HF) was added. The beaker was sealed and left on a hotplate at 100°C for a period of 24 hours. He found a negligible volume of rutile was dissolved. It should be noted that the glass in our study was preferentially dissolved by sonication in lower concentration of the acid (~0.6 M HF) at lower temperature (room temperature) for less time (~1.5 hours) than LPT by Wilkinson. (2015). It is thus not likely to cause collateral-side effects by rutile dissolution or leaching.

[12] Line 391: *"corresponding to $\Delta_{98/95}\text{Mo}_{\text{andesite-rutile}} = 0.90 \pm 0.17 \text{ ‰}$ at an appropriate 392 temperature for this study of 600°.* Top-slab temperatures at sub-arc depths are likely to be 700–900°C (Hermann and Spandler, 2008). Also, sediment melts seem almost always required to explain element transport across subduction zones. In the light of evidence for metasediments in the Raspas complex (Line 64), I think perhaps the authors need to better discuss the role of sediment melts in the light of their melt-involved experiments (see also Point 4 under Discussion below).

Response 12#:

The independent thermo-barometry on our samples (~600°C and ~2GPa) indicates they were not at the slab top. It is important to note that Herms et al. (2012) and Halama et al. (2011) document fluid flow structures and metasomatic effects caused by aqueous fluids whereas no evidences for hydrous melt or transitional fluids have been detected or reported for the Raspas Complex and Carbo Ortegá. The eclogite samples are themselves depleted MORB-like compositions and further indicate no role of sediment melt in their composition (see Fig. 1a).

In revision we have emphasized the lack of a sediment contribution in the samples we studied here.

Presentation

[13] It needs to be highlighted what an excellent documentation of analytical method and data quality regarding both the Mo isotope- and experimental work is presented here. The latter is comendable and the isotope group shows how it should be done.

One significant problem in this manuscript (+ supplement) is that insufficient geochemical background data for the samples are provided. This is in very stark contrast to the comendable point above.

Given the scope of the journal and the implications the authors suggest, neither a reviewer nor a reader should have to look up additional references for Ce (marked with citation in the table) to croscheck or build different diagrams. Moreover, Ba, Th, Nb, La etc... are not even cited although shown in diagrams. It is also important to know if for some samples such data (where not in diagrams) are unavaible or only for different sample splits etc. The latter would not be generally bad but nevertheless important to know. For instance Table 1 contains 10 eclogite/blueshist data and Fig. 1b only 8., so missing two which according to supporting literature "*Subducted seamounts in an eclogite-facies ophiolite sequence: the Andean Raspas Complex, SW Ecuador*" should be there (see Fig R1).

Currently such things are not transparent. This undermines the otherwise excellent work. I do not think that such practice should be accepted.

[13] Besides being a matter of comprehensiveness, there are also scientific considerations: Rutiles

from only one sample were analyzed and although the results seem convincing to support the main conclusion, I find it important to balance these limitations and have more comprehensive background geochemical data for the bulk samples to place them in the right context/to see if they are purely affected by fluid involvement (and not more complex hydrous melts). Some specific examples are given below.

Response 13#: We thank the reviewer for the kind comments and see the value in providing a comprehensive database to accompany the study. A data compilation of published major, trace elements and Sr-Nd isotopes has been attached as supplementary materials (Supplementary Table 1). Also, the Fig. 1b has been revised.

In revision we have provided a longer introduction to the general geochemistry of the samples from the literature data at the beginning of the manuscript.

Discussion

As much as I tend to agree with the model that rutile may be the complementary, light Mobearing mineral host compared to slab fluid-enriched mantle wedges, I would like to see the bulk samples better scrutinized.

[14] 1) What do bulk trace element systematics contribute to the issue of seawater contamination? In Line 95 it says that, "*Two serpentinites from the Raspas complex have high $\delta^{98/95}\text{Mo}_{\text{NIST 3134}}$ ($>0.1\%$), presumably reflecting their interaction with isotopically*

heavy seawater ($\delta^{98/95}\text{Mo}_{\text{NIST 3134}} \sim 2.3\%$)¹⁹ during serpentinisation. Although the serpentinites are enriched in Mo relative to Ce (Table 1), their Mo abundances are still very low (<20 ng/g) and markedly lower than even the most Mo-depleted eclogites.”

First, why then are these samples included in the eclogite suite, according to the Figure 1a legend, if they are serpentinites? In order to follow the authors explanation, where these two seawater contaminated samples plot in Fig. 1a (to see if slab fluid loss has clearly occurred according to Ba/Th) and why they show lower Mo/Ce (clearly a slab fluid-loss signature which leaves Mo heavy?).

Response 14#: The serpentinite analyses are included as a reference to illustrate that they have very low [Mo], <20ng/g and so will not have a major impact directly on the Mo budget in arc lavas. The presence of serpentinites in the Raspas complex affords us this opportunity for analysis. In general, however, we do not plot the serpentinites as their compositions are so different from the eclogites which are the focus of our attention. In light of the reviewer’s suggestion, we have plotted two serpentinites in to revised Fig. 1b (no available Nd isotope data for serpentinites). As can be seen, the two serpentinites have very high Ba/Th and Sr/Nd ratios, and also have high $\delta^{98/95}\text{Mo}_{\text{NIST3134}}$ and Mo/Ce, which reflects their interaction with isotopically heavy seawater during serpentinization. It should be noted that the serpentinites experienced de-serpentinization or slab fluid loss will have contrasting characteristics e.g., low Ba/Th, Sr/Nd, $\delta^{98/95}\text{Mo}_{\text{NIST3134}}$ and Mo/Ce. However, this kind of rock is unfortunately not sampled.

[15] 2) Further, starting in line 115 it says: “The latter observation implies that Ba is mobilised by fluids and removed from some samples, but local fluid addition can have an opposite effect”. If this is true should the repeated fluid removal and re-infiltration processes produce a trend in Ba/Th vs Mo isotopic composition? If it is less crucial whether blueschist or eclogite was present rather than the amount of fluid-rutile interaction and hence extent of reactive flow occurred, then the proxy for fluid scavenging or re-infiltration could show trends with the magnitude of fractionation.

However, it seems from Fig 1b that most of the Raspan samples have seen more fluids than the other samples investigated here. Yet these show the lowest Mo/Ce and Mo isotopic composition. Would not the opposite be expected from the authors statement above (Line 115)? Fluids that pass through the slab scavenge some Mo from rutile and are gone, thereby leaving less Mo where lower amounts of fluids remain? Why then are the samples with the lowest Mo/Ce the ones with the highest Ba/Th? This is at least counter intuitive and needs an explanation. I would therefore also be careful with accepting the statements in lines 116 to 121, which a reader cannot follow without the relevant trace element data.

Response 15#: We can see that we did not express this concept with enough clarity and thank the reviewer for pointing out the potential questions that arise. We have re-phrased these arguments in revision. In essence, the Mo systematics are robust to later fluid additions, in contrast to Ba, as there is no new phase that grows to incorporate Mo as there is for Ba (phengite). Thus late perturbation is only seen in the eclogites when two conditions are satisfied: 1) sufficient fluid has been added at a late stage (as reflected by high Ba/Th); 2) the samples were sufficiently depleted in Mo, for the late interacting fluid to have sufficient

leverage on the Mo budget of the rutile. The samples that are affected have both some of the highest Ba/Th and the lowest Mo/Ce. We hope that this is clearer in revision.

[16] 3) It is not properly discussed what the consequences are for that blueshist samples do not just show a typical N-MORB signature. In Line 67 it says: “*Most Raspas eclogites have geochemical characteristics indicative of a mid-ocean ridge basalt (MORB) protolith, whereas associated blueschists exhibit more enriched, seamountlike characteristics.*” Then in Line 212: “*There are still limited Mo isotope analyses of ocean island basalts (OIB) but $\delta^{98/95}\text{Mo}_{\text{NIST 3134}}$ lower than mid-ocean ridge basalt values are observed by Liang et al.42.*”

I note one red sample (notably a blueshist) that shows the lightest Mo isotopic composition. According to statement from line 212 this could then be a previously generated signature? Is this a chicken and egg issue? Well this sample certainly shows the highest Nb/La (at least I infer this but again do not have the actual data in a table). If so, then it is in principle ok to discuss it as affected by slab fluids (even twice...). However the quantification of the singular process (or relevance of Mo isotope value here) should be done without the blueschists. In any case this issue requires attention.

Response 16#: We agree that the discussion of blueschists has the potential to be a chicken and egg issue. However, the blueschist with the lowest $\delta^{98/95}\text{Mo}$ also has the most extreme Mo/Ce which is not a typical mantle signature, implying both are generated by fluid loss rather than differences in protolith. Moreover, in the admittedly limited current database, enriched MORB are isotopically heavy in Mo rather than isotopically light (Bezard et al., 2016) and so the sense of variation is opposite for this to be related to protolith. This point is now highlighted in the revised text.

[17] 4) It remains unclear why starting line 255 (Fig.2 legend) it says “arc lavas were selected with Ba/Th>400 to minimise complexity of variable sediment contributions”, when in Line 64 it says: “The Raspas Complex, Ecuador, contains metasediments...” Is this not contradictory? Why exactly is the dataset trimmed? What role do subducted sediments play in the system? Evidence exists for that hydrous melts are needed for efficient extraction of LILEs from subducted sediments (Hermann and Spandler, 2008). Can it be ruled out that sediment melts are involved in the former subduction setting and match hydrous melt-derived experimental results? Is the conclusion that merely fluids cause the observed signatures valid? Please keep in mind apparent discrepancies with Ba/Th pointed out above.

Response 17#: We think this is not contradictory.

1. The geochemical signature of arc lavas reflects variable contributions of sediment melts and fluid from the mafic oceanic crust. A global dataset contains samples with variable contributions of both. Subducting sediment can have a dominant control on the $\delta^{98/95}\text{Mo}$ of arc lavas (driving to isotopically heavy or light compositions depending on the subducting sediment type) (Freymuth et al., 2016; Freymuth et al., 2015; König et al., 2016; Wille et al., 2018) and can mask the signature from fluid that interacted with the mafic oceanic crust. Here we are interested in the latter and so focus only on arc lavas which are dominated by the contribution from this fluid. For this purpose, we filter for samples with Ba/Th>400.

2. Although there are metasediments as part of the Raspas complex, they are quite separate from the mafic eclogites we study and have not influenced their compositions. We have tried to make this clear in revision.

Further

[18] Line 56 what are typical oceanic sediments? Oceanic sediments display a large range of Mo isotopic composition from heavy signatures in reducing sediments, up to -2.1 permil down to -0.8 permil in deep sea pelagic sediments.

Response 18#: We see the problem here. Indeed, oceanic sediments are quite variable in their Mo isotopic composition. The point here was to exclude black shales (see Freymuth et al, 2016), which are rare and not typically involved in global arc volcanism. To try to avoid this issue we have changed 'typical' to 'most'.

[19] Line 106 Is this a combined effect of Mo (IV) bound octahedrally in rutile and MoO₄²⁻ as Mo (VI) in silica melt? Please specify!

Response 19#: This is an interesting point to which there is currently not a definitive answer. There is certainly a change in co-ordination, which is why we stress this. Whether or not there is also a change in redox during incorporation of Mo into rutile remains to be determined. Mo(VI) has an ionic radius closer to Ti(IV) and could be accommodated with appropriate charge balance for this minor component. However, it is possible that Mo is reduced during incorporation. This requires further work, so we don't speculate here.

[20] Line 363: "*No buffering system was applied to controloxygen fugacity, but previous experience with our piston cylinder has shown that the oxygen fugacity in such experiments buffers around Ni-NiO.*" This requires a bit more explanation and at least a reference but also an idea about the uncertainty involved

Response 20#: The imposed or 'intrinsic f_{O_2} ' control is a complicated issue that can take some time to design correctly and measure and is beyond the scope of this paper. As f_{O_2} was not a priority in these experiments, we went for the 'intrinsic' approach with no imposed buffer and a single capsule. As discussed at length in Matjuschkin et al. (2015), unbuffered experiments in various piston cylinder set ups, with excess water and Fe bearing melts, often experience 'water loss' or 'gain' as hydrogen moves to the most oxidized area, inside or outside the capsule which acts as a semi-permeable membrane $Fe^{3+/2+}$ changes to accommodate the change in amount of oxygen. In Bristol, the fluid loss tends to be minimal when the sample is loaded with a $Fe^{3+/2+}$ ratio close to that of the Ni-NiO buffer (± 1 log unit; Matjuschkin et al., 2015), so we assume this is the 'intrinsic' f_{O_2} of our assembly. As demonstrated by Matjuschkin et al. (2015), in cases where some solid-state buffer is used to counteract this and fight against a loaded $Fe^{3+/2+}$ ratio, these buffers tend to become exhausted after some amount of time except the Ni-NiO buffer, which again suggest this is 'backed up by the 'intrinsic' f_{O_2} . Similarly, unbuffered natural samples without water also tend to grow two-oxide pairs close to Ni-NiO or one log unit above, similar to the loaded rock material.

We are therefore confident that that the f_{O_2} of our experiments was approximately at Ni-NiO ± 1 log unit. Any variation near this oxygen fugacity is well within the realm of Mo^{6+} stability and should thus neither affect Mo speciation nor Mo isotope fractionation. We have summarised this discussion in lines 430-440 of the manuscript."

Reviewer #3 (Remarks to the Author):

[21] Subduction zones, plate margins where one tectonic plate is forced below the other and thus dragged into the Earth's mantle, are unique geodynamic features of Earth that keep it a living - and thus habitable - planet. Through the release of fluids (among which CO₂) back to the surface, subduction zones regulate the Earth's climate over geological timescales. The transport of cold crustal material into the hot Earth's mantle also plays an important role in maintaining planetary-scale mantle convection cells by which heat from the core-mantle boundary is transported to the Earth's surface. A clear understanding of the geological processes that operate in subductions zones is therefore vital to research fields far beyond those of Earth Sciences.

In this respect, this contribution adds a key element of information to this topical issue. It has long been thought that fluids released into the overlying arc volcanoes (and ultimately into the atmosphere) originate from the upper part of subducting plate. This part has been in contact with seawater during the million-years-long journey of tectonic plates from mid-ocean ridges, where they are formed by partial melting of the Earth's mantle, into the subduction zones, where they are recycled into the mantle as subducting slabs. Yet, although dehydration of fluids contained in the upper parts of subducting slabs may contribute to the volatile budget of arcs, its sheer volume appears to be too small.

Using state-of-the-art Mo isotope systematics, Chen and co-workers were able to draw a completely different picture of how slab dehydration may operate. Rather than a superficial, small-volume origin of fluids, their data lend important support for a model, whereby high-volume, water-rich mantle material underlying the cold (and dry) oceanic crust, releases its fluid-content at mantle depth, thus flushing the overlying (largely dry) slab with large amounts of supercritical fluids. Not only may this help to reconcile the volumes of fluids released into the volcanic arc but has also important implications for the chemical budget of the fluids that they are able to obtain during their passage through the chemically enriched subducting slab.

On top of that, the data also links up two important recent observations in isotope geochemistry: Volcanic rocks from subduction zones have an (overall) "heavy" Mo isotope composition, whereas mantle-derived rocks that potentially contain a recycled component have a "light" Mo isotope composition. The experiments and measurements by Chen and co-workers, for the first time, support a recently suggested hypothesis, that the dehydration and recycling process of tectonic plates in subduction zone is the driving force that caused these somewhat complimentary isotopic features.

The study itself has been meticulously designed and carried out. Both, experiments and isotope data adhere to the highest quality levels and there is little room for improvement in the overall assessment and interpretation of the data.

I have one minor comment that does not really affect the overall outcome of the study. In turn, I think that addressing it in a mildly revised version may help to strengthen the authors' case: The experiments have been carried out in Au-Pd capsules. Given that Mo is moderately siderophile and may thus form alloys with the capsules, it would be good to address this issue with respect to the partitioning (and thus fractionation) behaviour of Mo between the silicate melt and rutile. On the one hand, I reckon, the system will be heavily buffered by the high Mo concentration of the glass. On the other hand, only the relative isotopic difference between rutile and melt is of interest, not their absolute δMo values. Maybe this would be worth mentioning in the method section.

There are a few minor quibbles that are addressed below. Overall, I regard this manuscript as an excellent and novel contribution of potential interest to researchers beyond the field of Earth Sciences. It is well worth being published in Nature Communications following some (very) minor revisions.

Response 21#: We thank the reviewer for their recognition of both the extensive effort and value represented by our new datasets.

We have addressed the potential issue of Mo loss in line 465-469 of the Methods section. We do not think this has been an issue because the high oxygen fugacity of the experiments limits the affinity of Mo for metals. Moreover, as reviewer indicates, the Mo rich sample (effectively the glass, given its abundance and Mo content) buffers the system. A reconstructed bulk Mo content confirms this buffering assertion, because it yields indistinguishable Mo contents as the starting glass.

Detailed comments:

[22] Line 62: I reckon adding 'subducted' to oceanic lithosphere would emphasize the complementarity mentioned.

Response 22#: Revised.

[23] Lines 69 and 72: Second half of sentence sounds a bit odd. Maybe rephrase?

Response 23#: Revised.

[24] Line 109ff: This sentence is a bit confusing. How about: "We argue that the depletion of Mo in the eclogites compared to MORB, as recorded by their lower Mo/Ce, reflects the loss of Mo from the subducting crust to fluids that traverse the dehydrating slab. This is in keeping ...".

Response 24#: Thanks. We have rephrased here.

[25] Line 178: Wouldn't 'take up' be a better expression than 'dissolve'?

Response 25#: Thanks, but we think "dissolve" is more suitable here.

[26] Lines 186 and 187: Please clarify the nature of these experiments and observations. Isotopic, chemical, etc

Response 26#: The three studies we cited were chosen to encompass a wide range of different elemental and isotopic work used to infer oxygen fugacity. It is difficult to briefly summarise the diverse nature of these experiments so we prefer to leave just the references to the work.

[27] Lines 205 to 207: The statement “further down-dip of the paleo slab” seems to contradict the pressure estimates of the two breakdown reactions given (both at 2GPa).

Response 27#: This was a confusing statement. These reactions are rather insensitive to pressure and so the point had only been to contrast the reaction temperatures to those of the serpentine at peak metamorphic conditions. On reflection this does more to confuse than help and so the sentence has been rephrased.

[28] Line 288 (and below): Please change from ‘rock standard’ to ‘geological reference material’ (GRM) as neither W-2a nor any other reference material issued by the USGS or GSJ are standards.

Response 28#: Revised.

[29] Line 404: ‘Mo solubility in the minerals’ sounds odd. Do you mean Mo partitioning into (or between) minerals?

Response 29#: Yes, Mo solubility means Mo partitioning into minerals. We use this expression is in keeping with the reference (Bali et al.,2012) we cited.

References

- Bali, E., Keppler, H., Audetat, A., 2012. The mobility of W and Mo in subduction zone fluids and the Mo–W–Th–U systematics of island arc magmas. *Earth Planet. Sci. Lett.* 351-352, 195-207.
- Beinlich, A., Klemd, R., John, T., Gao, J., 2010. Trace-element mobilization during Ca-metasomatism along a major fluid conduit: Eclogitization of blueschist as a consequence of fluid–rock interaction. *Geochim. Cosmochim. Acta* 74, 1892-1922.
- Bezard, R., Fischer-Gödde, M., Hamelin, C., Brennecke, G.A., Kleine, T., 2016. The effects of magmatic processes and crustal recycling on the molybdenum stable isotopic composition of Mid-Ocean Ridge Basalts. *Earth Planet. Sci. Lett.* 453, 171-181.
- Bloch, W., John, T., Kummerow, J., Salazar, P., Krüger, O.S., Shapiro, S.A., 2018. Watching dehydration: Seismic Indication for Transient Fluid Pathways in the Oceanic Mantle of the Subducting Nazca Slab. *Geochem. Geophys. Geosyst.*
- Debret, B., Sverjensky, D.A., 2017. Highly oxidising fluids generated during serpentinite breakdown in subduction zones. *Sci. Rep.* 7, 10351.
- Freyruth, H., Elliott, T., van Soest, M., Skora, S., 2016. Tracing subducted black shales in the Lesser Antilles arc using molybdenum isotope ratios. *Geology*, G38344.38341.

- Freytmuth, H., Vils, F., Willbold, M., Taylor, R.N., Elliott, T., 2015. Molybdenum mobility and isotopic fractionation during subduction at the Mariana arc. *Earth Planet. Sci. Lett.* 432, 176-186.
- Gao, J., John, T., Klemd, R., Xiong, X., 2007. Mobilization of Ti–Nb–Ta during subduction: Evidence from rutile-bearing dehydration segregations and veins hosted in eclogite, Tianshan, NW China. *Geochim. Cosmochim. Acta* 71, 4974-4996.
- Halama, R., John, T., Herms, P., Hauff, F., Schenk, V., 2011. A stable (Li, O) and radiogenic (Sr, Nd) isotope perspective on metasomatic processes in a subducting slab. *Chem. Geol.* 281, 151-166.
- Hermann, J., Spandler, C., Hack, A., Korsakov, A.V., 2006. Aqueous fluids and hydrous melts in high-pressure and ultra-high pressure rocks: implications for element transfer in subduction zones. *Lithos* 92, 399-417.
- Herms, P., John, T., Bakker, R.J., Schenk, V., 2012. Evidence for channelized external fluid flow and element transfer in subducting slabs (Raspas Complex, Ecuador). *Chem. Geol.* 310-311, 79-96.
- John, T., Gussone, N., Podladchikov, Y.Y., Bebout, G.E., Dohmen, R., Halama, R., Klemd, R., Magna, T., Seitz, H.-M., 2012. Volcanic arcs fed by rapid pulsed fluid flow through subducting slabs. *Nat. Geosci.* 5, 489-492.
- John, T., Scherer, E.E., Haase, K., Schenk, V., 2004. Trace element fractionation during fluid-induced eclogitization in a subducting slab: trace element and Lu–Hf–Sm–Nd isotope systematics. *Earth Planet. Sci. Lett.* 227, 441-456.
- John, T., Scherer, E.E., Schenk, V., Herms, P., Halama, R., Garbe-Schönberg, D., 2010. Subducted seamounts in an eclogite-facies ophiolite sequence: the Andean Raspas Complex, SW Ecuador. *Contrib. Mineral. Petrol.* 159, 265-284.
- König, S., Wille, M., Voegelin, A., Schoenberg, R., 2016. Molybdenum isotope systematics in subduction zones. *Earth Planet. Sci. Lett.* 447, 95-102.
- Kessel, R., Schmidt, M.W., Ulmer, P., Pettke, T., 2005. Trace element signature of subduction-zone fluids, melts and supercritical liquids at 120-180 km depth. *Nature* 437, 724-727.
- Matjuschkin, V., Brooker, R.A., Tattitch, B., Blundy, J.D., Stamper, C.C., 2015. Control and monitoring of oxygen fugacity in piston cylinder experiments. *Contrib. Mineral. Petrol.* 169.
- Plümper, O., John, T., Podladchikov, Y.Y., Vrijmoed, J.C., Scambelluri, M., 2017. Fluid escape from subduction zones controlled by channel-forming reactive porosity. *Nat. Geosci.* 10, 150-156.
- Spandler, C., Pirard, C., 2013. Element recycling from subducting slabs to arc crust: A review. *Lithos* 170, 208-223.
- Taetz, S., John, T., Bröcker, M., Spandler, C., Stracke, A., 2018. Fast intraslab fluid-flow events linked to pulses of high pore fluid pressure at the subducted plate interface. *Earth Planet. Sci. Lett.* 482, 33-43.
- Voegelin, A.R., Pettke, T., Greber, N.D., von Niederhäusern, B., Nägler, T.F., 2014. Magma differentiation fractionates Mo isotope ratios: Evidence from the Kos Plateau Tuff (Aegean Arc). *Lithos* 190-191, 440-448.

- Wilkinson, D.J., 2015. Geochemistry of eclogites from Western Norway: implications from high-precision whole-rock and rutile analyses. University of Edinburgh.
- Willbold, M., Elliott, T., 2017. Molybdenum isotope variations in magmatic rocks. *Chem. Geol.* 449, 253-268.
- Wille, M., Nebel, O., Pettke, T., Vroon, P.Z., König, S., Schoenberg, R., 2018. Molybdenum isotope variations in calc-alkaline lavas from the Banda arc, Indonesia: Assessing the effect of crystal fractionation in creating isotopically heavy continental crust. *Chem. Geol.* 485, 1-13.

Reviewers' comments:

Reviewer #2 (Remarks to the Author):

Molybdenum systematics of subducted crust record extensive, reactive flow from underlying slab-serpentine dehydration

Some major points raised in the former round of revision were adequately addressed. In particular the relevance of the experiments is more discussed, the various fluid infiltration- and dehydration events are attempted and the negligible sediment role in Raspas eclogite petrogenesis is better explained. I also appreciate the incorporation of relevant geochemical data in a table to the supplement of the revised manuscript.

The novelty of this study needed to be better worked out to justify publication in a nature journal. The authors worked on this part too starting with a slightly changed title, modified introduction L48 to 64 and a new cartoon (Fig. 4), all in order to scrutinize aspects of underlying serpentinite dehydration vs. overlying serpentinite dehydration and melange-afflicted scenarios. I find that this manuscript significantly contributes arguments in favour of the first scenario and therefore deserves consideration for publication in Nature Communications.

I recommend this consideration only after some remaining, critical issues are properly addressed. These are more related to the details rather than the main argument but still need to be clearer. Points below are ordered by appearance in the manuscript and not by relevance.

Abstract

I still find the abstract is lagging behind the above mentioned improvements and should be clearer, most importantly for a more general readership. Mo isotopes are not needed to trace the process of slab devolatilization (this can be achieved by trace elements or radiogenic Pb) rather than to further investigate the more detailed mechanisms and even relevance of slab stratigraphy (serpentinites underlying rutile-bearing oceanic crust). To me it also reads like the process is required to explain the Mo isotopes rather than the novel tracer now clearly identifies the process. The authors should also mention that experiments were done and together with the newly investigated HP rocks refine the tracer. The last sentence includes an observation about Mo isotopes, rather than a concluding remark about a process!

Experiments

The coordination argument is plausible as such and I agree that the direction of Mo isotope fractionation is reproduced, which should be sufficient here to make the main point. Still, physical conditions of melts vs. fluids are very different and it should not say in the main text (L152) that "fluid phase is isotopically heavier than the rutile"...rather than liquid phase or sth....

Also in the supplement experiments are mentioned to contain ca. 14% rutile (which is a lot compared to ca 1.4% assumed for their later model L204). Do I miss sth here? Is this needed to cause this effect? How does this relate to nature? How reliable then are subsequent quantifications? What would happen with much less rutile in the experiments? I note that the relatively high FMQ of +5 has been critically mentioned before. Indeed it is way above the upper one calculated by Bali et al in the cited study. Is there room for moderation?

Finally, I note in the literature that melts from rutile-bearing eclogite are shown to contain way different inter-HFSE ratios (like Nb/Ta), which are not known to be caused by fluids. Is it possible to extrapolate from the hydrous melt experiments the effect of lower melt-to-hydrous proportion? I know the effect should not disappear but could this perhaps lead to more moderate FMQ? I simply think the current quantification might be better described as tentative.

Discussion

Late re-enrichment of Ba but not Mo

I understand here that, after initial Mo loss only Ba is later re-enriched because the retention potential of the overlying crust is exhausted. This in turn is the case because later fluids are Ti-poor and thus rutile does not anymore grow (only phases like phengite that trap Ba, Rb etc). Why are these fluids Ti-poor and why is the Ti from MORB (L205) not anymore sufficient to account for further Mo-retaining rutile?

In the following paragraph it is mentioned though that some Mo is re-enriched to explain the two eclogite samples that fall off the trend (see also L. So is this fluid now Ti-rich or is there simply rutile present? How can it be reconciled with the previous part?

Fig 2 can also be interpreted as showing that with decreasing Mo/Ce after 0.004 the isotope variability dramatically increases (4 samples) because at such low relative Mo abundances, any additional loss or re-enrichment may have "dramatic" effects.

Is this not a more likely explanation? In any case I am reluctant to over-interpret the two lowest $d_{98}\text{Mo}$ -bearing samples as well as the two that fall off the trend, because all other samples with Mo/Ce above 0.004 are already complementary to arc lavas. perhaps the authors should focus more on explaining most of their samples rather than some extreme values (L214 and following is an example) which are then translated into extreme models. This is not necessary and weakens the manuscript.

Model

I noted earlier that for the model only 1.4% rutile is taken which is by factor 10 less than used in the experiments. Experimental and modeling parameters should be the same or not?

Evidence for external fluid and composition of this fluid

This is an important issue but one of the arguments presented (that blueschists show the most extreme values) is not correct. Only one of the most extreme samples is a blueschist, the others all eclogites. See also comment on Fig 2 above.

L236 and following...origin of Mo...Freymuth et al. (2016) (also L273) argued for a lower AOC origin of Mo...I do not understand the sentences here in that regard. Please rephrase.

Following the Freymuth et al and in L278 repeated arguments, Mo also comes from deeper AOC. If the source of fluids is only serpentinite (as argued L278), how can the deeper (one order of magnitude more required) Mo come up? This is still unclear.

Cartoon (final figure)

The cartoon completely leaves out the deeper AOC origin of Mo (L278) and that the fluids signature is changed. The caption states the latter but why then have a cartoon? This can be improved!

Additional Line by Line:

32: A large (?) fluid....

38: Superlatives are always tricky. Along MOR significantly more melt volume is produced and interacts with surface seawater. So it could be argued that MOR are the primary sites of chemical exchange no?...perhaps rephrase to say that they are the "principal sites of surface material recycling into the mantle"

L258 odd sentence...remove from?

L287 odd sentence

Significant figures in all tables would be nice

Adverb for the title instead of extensive (?)

Blueschists should be blue symbols in the figures to avoid confusion

Reviewer #3 (Remarks to the Author):

In the revised version of this manuscript as well as in their detailed replies, the authors have adequately and comprehensively addressed the comments raised by the reviewers. The concerns flagged were mostly appropriate and justified comments never concerned issues that would have weakened or even invalidated the basic interpretations or conclusions of the manuscript. Some of the more critical concerns were - to some extent - based on slight misunderstandings on the side of the reviewers, which, nevertheless makes these comments valid given that other readers might have shared the same views hadn't these been flagged during the review process. The revised manuscript now avoids such misconceptions to occur.

Thus, the revised version of the manuscript, in my opinion, does justice to all the points raised by the reviewers while retaining the original key messages and the major thrust of this contribution. In the end, the authors took full advantage of the comments raised during the review process to strengthen their arguments and to present their interpretations in an even more comprehensive way in the revised manuscript.

I also believe, that the authors now fully comply with making available the full dataset, as requested by Reviewer No.2.

I reiterate my previous assessment that this contribution is highly original and topical and that its impact will have ramifications beyond Earth Sciences. Following this very thorough and comprehensive review process I therefore highly recommend publication of the manuscript in its present form.

Reviewer 1#

Re□Review of Nature Comm

Overall, I am pleased that the authors have toned down their statements that arc lavas are heavy because of slab□derived fluids only. They remain with their statement in the text that this is the key cause for the heavy Mo Isotopes in arc lavas – and I still think this is a bold statement. However, I respect their interpretation. Peer review is not and should not be any form of scientific censorship and this is why we distinguish between results and discussion □ so their interpretation should go out there and stand up for a scientific disputation. This work is scientifically sound, and I am sure it will advance science; the one or the other way

The only issue I have really is that the statement that fractionation cannot account for heavier Mo isos, even in more primitive arc lavas, is imho not substantiated by the data here or any other data elsewhere (see below). This, however, is not really the crux here and I can clearly see how this paper can add to a better understanding of dehydration processes in arcs.

I suggest publication of this work with minor revisions now □ even though I still disagree with parts of their interpretation. □ We can agree to disagree here. I truly believe this is how science advances.

We thank the reviewer for his positive view of our latest revisions.

Some comments to consider

[1] The issue of magmatic differentiation

Citation of Willbold and Elliott. Stating that a citation of this work covers most arguments against magmatic arc differentiation as the cause for the heavy isotopes is not really sufficient. As the authors know (since Tim Elliott is a co□author here, too), Willbold and Elliott did not rule out that the magmatic rocks at Kos are affected by fractional crystallisation. The step□wise move outlined in their review could may well be a statistical issue and/or still be caused by fractionation – or not. However, this is not a counter□argument here. The other papers cited conclude that fluid mobility is key, yet again there is no convincing argument in here. I cannot see any plot of Mo isotopes vs a fluid tracer, be that Ba/Th or Sr or Pb isotopes that indicates this.

I cite Willbold and Elliott : *In these best□case examples, the isotopic differences between most mafic and evolved endmembers can be reasonably reproduced, but the intermediate lavas require a more complex petrogenesis. In all, this suggests that the range in $\delta^{98/95}\text{Mo}$ values observed in the Kos sample set may have been caused by fractional crystallisation of hydrous phases but the involvement of genetically different magmatic systems or melt batches cannot be ruled out. As we discuss below, subduction zone processes can result in notable variability in $\delta^{98/95}\text{Mo}$ and so distinguishing primary from secondary influences on $\delta^{98/95}\text{Mo}$ in the Kos lavas is important but currently unresolved.*

And of course this boils down to the question where and how FC is affecting arc lavas. Looking into recent literature (e.g., Jackson et al is a good example <https://www.nature.com/articles/s41586□018□0746□2>) gives some insights into the

complicated nature of this and how prx□amph fract combined can do the trick. In this scenario, lower arc crust differentiation in a feasible alternative scenario to simple FC would add to an increase in Mo isotopes, but also allow for some scatter here. In fact, a simple “fractionation line” is not really what one expects even within single volcanoes, let alone on a global level. Even considering that this process would result in scatter in a plot of Mo isotopes vs SiO₂, it is still remarkable that there is a trend and this, in my opinion, can only be created by magmatic differentiation.

Now bottom line is that Willbold and Elliot give no evidence against FC as the cause for the heavy Mo isotopes. A citation of this review paper does not fix the issue here and I concur that at the very least, both scenarios are feasible.

Furthermore, looking into the Banda arc data – The exclusion of the 2 light data points (there were 4 btw) in Wille et al was done on a sound basis of O isotopes that indicate sediment/arc crust assimilation. Just adding them back to the data and making an average trend is deceptive at best or simply wrong – and using an R2 without taking error bars into account is also pretty strange. Alas □ taking all arc data together (admittedly the Martinique data are the odd ones out towards negative values but similar to the boninites this is sub sediment, right?), one can absolutely see a trend, ranging from MORB□like values at □0.2 and ca. 47 wt.% SiO₂ towards near zero values at 55 wt.% SiO₂ and a further increase towards +0.2 in the more evolved data. This does include primitive lavas. Changing the scale here would certainly also help with a starting well above □1.2. □ Saying there is none doesn't really make the trend go away, as much as grass remains green even if one says otherwise. It is totally correct that this is not the nicest of trends – I've seen better in my days, but still. We are looking for co□variations, not correlations, as evidence for a process driven effect (likely amphibole fractionation through reactive incongruent melting in a percolating flow environment or in other words through melt□rock interaction in lower arc crust) and not a perfectly calculated fractional crystallization sequence. What we are looking at here is a liquid line of descent in a global dataset. Wille et al., made the point clear that taken the Banda data in isolation fails, yet in a global dataset one can clearly see the rise of Mo isotopes with SiO₂. ... So do the authors really think that all of these data□points are elevated in their Mo isotopes by the right addition of fluids to create what appears to be a fractionation trend? I certainly accept that fluid can add to the noise in the data but that is as far as I go.

Response 1#: We appreciate the fair-minded debate sparked by the reviewer and we try to reciprocate in similar fashion. We would agree that the effect of magmatic differentiation on Mo isotope fractionation remains debatable. It is also true that the arguments of Willbold and Elliott (2017) do not rule out a role of fractional crystallization as a means of increasing $\delta^{98/95}\text{Mo}$ in more evolved samples. Thus, we have modified our mention of this reference to ‘see additional discussion in ref.⁵⁷’ rather than ‘see counter arguments in ref.⁵⁷’. For the purposes of this study, however, the key point is that mafic, sediment poor arc lavas have high $\delta^{98/95}\text{Mo}$. So regardless of the effects of differentiation, this needs to be explained. We tried to emphasize this in revision but also now more clearly leave open the possibility of other processes occurring during differentiation with the modification above.

[2] The Ba/Th argument – and boninites.

Now I went back and had a good look at Stefan Koenigs paper. Indeed I misstated here that the boninites (or high-Mg andesites for that matter) are heavy – they are indeed light. Mea Culpa.

Now all of the samples in that study are plotting in the range of MORB composition – or are lighter. So whatever fluid was added to these boninites (they may not be all pure boninites and sediment was involved) – this did not lift their Mo isotopes beyond MORB levels. Further looking at Koenigs Figure 3A – all of the literature samples plotted with heavier than MORB Mo isotopes range from low Ba/Th to high Ba/Th. Wouldn't one expect a systematic shift here if fluids all do that ?

Response 2#: Fundamentally we argue that a fluid derived by slab serpentine dehydration that interacts with the overlying mafic crust acquires high Ba/Th and high $\delta^{98/95}\text{Mo}$. Whether or not there is a clear correlation between these two parameters in arc lavas containing a fluid component depends on the simplicity of the arc system. König et al. (2016) presents data for a range of different magmas (boninites, boninitic back arc basalts, high magnesium andesites and a few basalts) from complex arc settings. Despite the marked petrogenetic diversity in this sample set, König et al. (2016) point out that they are broadly consistent with mixing between a high Ba/Th and high $\delta^{98/95}\text{Mo}$ fluid endmember and a low Ba/Th and low $\delta^{98/95}\text{Mo}$ sedimentary component. It might alternatively be viewed as surprising that such heterogeneous sample set shows an overall trend defined by such a simple model. Perhaps a more appropriate test of the model is the $\delta^{98/95}\text{Mo}$ found in other, depleted, high Ba/Th arc basalts. An abstract in the ongoing Goldschmidt conference for samples from the Izu arc reports that these samples indeed have $\delta^{98/95}\text{Mo}$ higher than MORB (Villalobos-Orchard et al., 2019).

[3] Oxidation.

I am still suspicious. The sulphide-sulphate transition at zero pressure (let's assume the same delta FMQ applies at depth) is at FMQ+2 or below. Sulphur has no stronger oxidising capacity. So...which is the magic oxidiser here – C? Or can the salinity enhance Mo mobilisation even at lower $f\text{O}_2$?

Response 3#: We emphasise that the range of $f\text{O}_2$ we consider are not *ad hoc* choices but values taken from recent peer-reviewed literature. Rather than inexpertly paraphrase a detailed body of work, we point the reviewer towards the manuscripts of Baptiste Debret, who has published several observational and theoretical studies to argue for such highly oxidized fluids liberated from the slab. In our manuscript we cite the calculations of Debret and Sverjensky (2017), but perhaps more tangibly Debret et al. (2015) report haemetite-magnetite intergrowths formed during deserpentinisation of the Cerro del Almirez massif, which would buffer fluids released to $f\text{O}_2$ to at least $\sim\text{FMQ}+3$.

In addition, recent experiments have demonstrated that the eight-electron transition from sulphate to sulphide is highly pressure dependant and easily shifted these high $f\text{O}_2$ values above 2 GPa (Matjuschkin et al., 2016). We cited this reference in

revision.

Oliver Nebel, Melbourne, June 17th 2019

Reviewer #2 (Remarks to the Author):

Molybdenum systematics of subducted crust record extensive, reactive flow from underlying slab-serpentine dehydration

Some major points raised in the former round of revision were adequately addressed. In particular the relevance of the experiments is more discussed, the various fluid infiltration- and dehydration events are attempted and the negligible sediment role in Raspas eclogite petrogenesis is better explained. I also appreciate the incorporation of relevant geochemical data in a table to the supplement of the revised manuscript.

The novelty of this study needed to be better worked out to justify publication in a nature journal. The authors worked on this part too starting with a slightly changed title, modified introduction L48 to 64 and a new cartoon (Fig. 4), all in order to scrutinize aspects of underlying serpentinite dehydration vs. overlying serpentinite dehydration and melange-afflicted scenarios. I find that this manuscript significantly contributes arguments in favour of the first scenario and therefore deserves consideration for publication in Nature Communications.

I recommend this consideration only after some remaining, critical issues are properly addressed. These are more related to the details rather than the main argument but still need to be clearer. Points below are ordered by appearance in the manuscript and not by relevance.

We thank the reviewer for their positive view of our latest revisions. We respond to the reviewer queries mainly by minor clarifications and changes both in this response and within the body of the manuscript.

[4] Abstract

I still find the abstract is lagging behind the above mentioned improvements and should be clearer, most importantly for a more general readership. Mo isotopes are not needed to trace the process of slab devolatilization (this can be achieved by trace elements or radiogenic Pb) rather than to further investigate the more detailed mechanisms and even relevance of slab stratigraphy (serpentinite s underlying rutile-bearing oceanic crust). To me it also reads like the process is required to explain the Mo isotopes rather than the novel tracer now clearly identifies the process. The authors should also mention that experiments were done and together with the newly investigated HP rocks refine the tracer. The last sentence includes an observation about Mo isotopes, rather than a concluding remark about a process!

Response 4#: We tried to modify the abstract following the reviewer's suggestions.

We have removed reference to Mo as a tracer. We have also highlighted the experiments we undertook. However, we wish to retain the final sentence of broader implication of our work, which is in keeping with the Nature Communications outline of an abstract.

[5] Experiments

The coordination argument is plausible as such and I agree that the direction of Mo isotope fractionation is reproduced, which should be sufficient here to make the main point. Still, physical conditions of melts vs. fluids are very different and it should not say in the main text (L152) that “fluid phase is isotopically heavier than the rutile”...rather than liquid phase or sth....

Response 5#: Revised.

[6] Also in the supplement experiments are mentioned to contain ca. 14% rutile (which is alot compared to ca 1.4% assumed for their later model L204). Do I miss sth here? Is this needed to cause this effect? How does this relate to nature? How reliable then are subsequent quantifications? What would happen with much less rutile in the experiments? I note that the relatively high FMQ of +5 has been critically mentioned before. Indeed it is way above the upper one calculated by Bali et al in the cited study. Is there room for moderation?

Response 6#: The fractionation factor between melt/fluid and rutile is not dependent on the amount of rutile (as long as equilibrium is obtained, which we infer to be the case). An experiment with a relatively high proportion of rutile is experimentally desirable as it needs to yield sufficient rutile to analyse.

Moreover, as we show in the supplementary material and flag in the text, the slope of the sample array in Fig. 2 is essentially controlled by the fractionation factor. Hence our experimental measurement is a valuable consistency check, but the fractionation factor used in modelling is pinned by the observations.

Bali et al. (2012) conducted experiments at a range of oxygen buffers, but admittedly only as oxidized as Re-ReO₂. Nonetheless, the experiments were parameterised to allow extrapolation to higher fO_2 . Although imperfect, this approach still provides a valuable estimate. It would be good to have new experiments covering more oxidized conditions, but this is not likely in the near future. However, there is room for moderation and we show in Fig. 3 solutions for values from FMQ+3 to +5.

[7] Finally, I note in the literature that melts from rutile-bearing eclogite are shown to contain way different inter-HFSE ratios (like Nb/Ta), which are not known to be caused by fluids. Is it possible to extrapolate from the hydrous melt experiments the effect of lower melt-to-hydrous proportion? I know the effect should not disappear but could this perhaps lead to more moderate FMQ? I simply think the current quantification might be better described as tentative.

Response 7#: This is an interesting suggestion which can be tried in the future but is not possible with experimental data to hand. We would agree with the reviewer that current experimental characterisation of the elemental and isotopic fractionations of

Mo during slab dehydration is imperfect, but the work of Bali et al. (2012) makes Mo partitioning between slab and fluid rather better quantified than for than many elements. For most elements any data on isotopic fractionation in this scenario is entirely lacking. Hopefully this dearth of data will be addressed in coming years, but at present the situation for Mo is comparatively well constrained.

Discussion

[8] Late re-enrichment of Ba but not Mo

I understand here that, after initial Mo loss only Ba is later re-enriched because the retention potential of the overlying crust is exhausted. This in turn is the case because later fluids are Ti-poor and thus rutile does not anymore grow (only phases like phengite that trap Ba, Rb etc). Why are these fluids Ti-poor and why is the Ti from MORB (L205) not anymore sufficient to account for further Mo-retaining rutile?

In the following paragraph it is mentioned though that some Mo is re-enriched to explain the two eclogite samples that fall off the trend (see also L. So is this fluid now Ti-rich or is there simply rutile present? How can it be reconciled with the previous part?

Fig 2 can also be interpreted as showing that with decreasing Mo/Ce after 0.004 the isotope variability dramatically increases (4 samples) because at such low relative Mo abundances, any additional loss or re-enrichment may have “dramatic” effects. Is this not a more likely explanation? In any case I am reluctant to over-interpret the two lowest $d_{98}\text{Mo}$ -bearing samples as well as the two that fall off the trend, because all other samples with Mo/Ce above 0.004 are already complementary to arc lavas. perhaps the authors should focus more on explaining most of their samples rather than some extreme values (L214 and following is an example) which are then translated into extreme models. This is not necessary and weakens the manuscript.

Response 8#: The fluid is Ti-poor because under eclogite conditions rutile is stable and this significantly limits the [Ti] in the fluid.

The two anomalous samples are still Mo poor (low Mo/Ce) because no new rutile grows. However, if the existing rutile equilibrates with isotopically heavy Mo in the late introduced fluid, the rutile will acquire a more elevated $\delta^{98/95}\text{Mo}$ (as found for the rutile in SEC43-1) and thus plot off the main trend.

We agree that small additions of fluid to low [Mo] eclogites can have dramatic effects. However, since the fluid will have isotopically heavy Mo, late fluid addition should result in a direction of change to the top and right, not a vertical vector as implied by the sample distribution in Fig 3. In general, we agree that these two samples are a distraction. We felt we should try to explain them, but do not want this to dominate. Given the comments of the reviewer, we have greatly trimmed out reference and explanation of these samples, which we hope highlights our thoughts less intrusively on the main discussion.

[9] Model

I noted earlier that for the model only 1.4% rutile is taken which is by factor 10 less than used in the experiments. Experimental and modeling parameters should be the same or not?

Response 9#: As discussed in Response 6#, an equilibrium experiment gives the fractionation factor independent of the amount of rutile present. This fractionation factor then does have to be combined in appropriate modal proportion for an eclogite with the contributions of omphacite and garnet (as determined from inter-mineral fractionation measurements on sample SEC46-1) to yield a bulk eclogite fractionation factor. This is what we have done.

[10] Evidence for external fluid and composition of this fluid

This is an important issue but one of the arguments presented (that blueschists show the most extreme values) is not correct. Only one of the most extreme samples is a blueschist, the others all eclogites. See also comment on Fig 2 above.

Response 10#: We have revised this part. Please see response 8#.

[11] L236 and following...origin of Mo...Freymuth et al. (2016) (also L273) argued for a lower AOC origin of Mo...I do not understand the sentences here in that regard. Please rephrase.

Response 11#: There are two arguments presented in Freymuth et al. (2015). One is a mass balance. Given the upper, altered crust has high $\delta^{98/95}\text{Mo}$, the high $\delta^{98/95}\text{Mo}$ in arc lavas might be argued to derive from this reservoir. However, there is insufficient Mo in this portion of the crust to supply enough isotopically heavy Mo given crustal growth rates and arc lavas [Mo]. The second argument is that the upper portion of altered, old Pacific mafic crust that subducts beneath the Marianas has a distinctive Pb isotopic composition (resulting from decay of U added near the ridge). The tightly defined Pb isotope arrays of the Mariana arc lavas extrapolate to a fluid component akin to fresh Pacific mid-ocean ridge basalts and far from the distinctive altered oceanic crust. Thus, they infer that the Pb and Mo budgets of the fluid component are dominated by contributions from the deeper, less altered (cooler) portion of the subducting Pacific crust.

We can see that these points were rather cryptic in the original and we have now hopefully made them clearer (Please see Lines 262-269 in revision).

[12] Following the Freymuth et al and in L278 repeated arguments, Mo also comes from deeper AOC. If the source of fluids is only serpentinite (as argued L278), how can the deeper (one order of magnitude more required) Mo come up? This is still unclear.

Response 12#: We can address this question from two sides.

The Mo flux is contributed by Mo partitioned into the fluid the unaltered mafic crust, which has a stratigraphic thickness ~6500m, compared to the 500m of upper, altered crust. Although the upper, altered crust has isotopically heavy Mo due to exchange with seawater, it is not elementally enriched in Mo. Thus the difference in thickness of the altered and unaltered crustal unit account for the order of magnitude difference in

available Mo.

[13] Cartoon (final figure)

The cartoon completely leaves out the deeper AOC origin of Mo (L278) and that the fluids signature is changed. The caption states the latter but why then have a cartoon? This can be improved!

Response 13#: Thank for the comment, we have tried to modify this accordingly.

Additional Line by Line:

[14] 32: A large (?) fluid....

Response 14#: We modified here.

[15] 38: Superlatives are always tricky. Along MOR significantly more melt volume is produced and interacts with surface seawater. So it could be argued that MOR are the primary sites of chemical exchange no?...perhaps rephrase to say that they are the “principal sites of surface material recycling into the mantle”

Response 15#: Revised.

[16] L258 odd sentence...remove from?

Response 16#: Revised.

[17] L287 odd sentence

Response 17#: Revised.

[18] Significant figures in all tables would be nice

Response 18#: Revised.

[19] Adverb for the title instead of extensive (?)

Response 19#: Revised.

[20] Blueschists should be blue symbols in the figures to avoid confusion

Response 20#: We changed the color of the symbols as the review suggested.

Reviewer #3 (Remarks to the Author):

In the revised version of this manuscript as well as in their detailed replies, the authors have adequately and comprehensively addressed the comments raised by the reviewers. The concerns flagged were mostly appropriate and justified comments never concerned issues that would have weakened or even invalidated the basic interpretations or conclusions of the manuscript. Some of the more critical concerns were - to some extent - based on slight misunderstandings on the side of the reviewers, which, nevertheless makes these comments valid given that other readers might have shared the same views hadn't these been flagged during the review process. The revised manuscript now avoids such misconceptions to occur.

Thus, the revised version of the manuscript, in my option, does justice to all the points raised by the reviewers while retaining the original key messages and the major thrust of this contribution. In the end, the authors took full advantage of the comments raised during the review process to strengthen their arguments and to present their interpretations in an even more comprehensive way in the revised manuscript.

I also believe, that the authors now fully comply with making available the full dataset, as requested by Reviewer No.2.

I reiterate my previous assessment that this contribution is highly original and topical and that its impact will have ramifications beyond Earth Sciences. Following this very thorough and comprehensive review process I therefore highly recommend publication of the manuscript in its present form.

We thank the reviewer for the positive assessment regarding the significance and novelty of our study

References

- Bali, E., Keppler, H., Audetat, A. (2012) The mobility of W and Mo in subduction zone fluids and the Mo–W–Th–U systematics of island arc magmas. *Earth Planet. Sci. Lett.* **351-352**, 195-207.
- Debret, B., Bolfan-Casanova, N., Padrón-Navarta, J.A., Martin-Hernandez, F., Andreani, M., Garrido, C.J., López Sánchez-Vizcaíno, V., Gómez-Pugnaire, M.T., Muñoz, M., Trcera, N. (2015) Redox state of iron during high-pressure serpentinite dehydration. *Contrib. Mineral. Petrol.* **169**.
- Debret, B., Sverjensky, D.A. (2017) Highly oxidising fluids generated during serpentinite breakdown in subduction zones. *Sci. Rep.* **7**, 10351.
- Freymuth, H., Vils, F., Willbold, M., Taylor, R.N., Elliott, T. (2015) Molybdenum mobility and isotopic fractionation during subduction at the Mariana arc. *Earth Planet. Sci. Lett.* **432**, 176-186.
- König, S., Wille, M., Voegelin, A., Schoenberg, R. (2016) Molybdenum isotope systematics in subduction zones. *Earth Planet. Sci. Lett.* **447**, 95-102.
- Matjuschkin, V., Blundy, J.D., Brooker, R.A. (2016) The effect of pressure on sulphur speciation in mid- to deep-crustal arc magmas and implications for the formation of porphyry copper deposits. *Contrib. Mineral. Petrol.* **171**.
- Villalobos-Orchard J, Freymuth H, O'Driscoll B, Elliott T, Williams H, Casalini M & Willbold M. (2019) Tracing Subduction Zone Fluids in Izu Arc Lavas Using Molybdenum Isotopes. Goldschmidt2019 Abstract.
- Willbold, M., Elliott, T. (2017) Molybdenum isotope variations in magmatic rocks. *Chem. Geol.* **449**, 253-268.

REVIEWERS' COMMENTS:

Reviewer #2 (Remarks to the Author):

The authors have answered my previous questions and adequately addressed all remaining issues. I appreciate the time and effort taken to consider so many aspects. Now this work will be a valuable contribution that I recommend for publication in Nature Communications in its present form.

Reviewer #2 (Remarks to the Author):

The authors have answered my previous questions and adequately addressed all remaining issues. I appreciate the time and effort taken to consider so many aspects. Now this work will be a valuable contribution that I recommend for publication in Nature Communications in its present form.

Response: We highly appreciate the reviewer for the supportive comments.